# Microphysical process of precipitating hydrometeors from warm-front mid-level stratiform clouds revealed by ground-based lidar observations

Yang Yi[1,2,3], Fan Yi[1,2,3,*], Fuchao Liu[1,2,3], Yunpeng Zhang[1,2,3],Changming Yu[1,2,3], and Yun He[1,2,3]

[1]School of Electronic Information, Wuhan University, Wuhan 430072, China

[2]Key Laboratory of Geospace Environment and Geodesy, Ministry of Education, Wuhan 430072, China

[3]State Observatory for Atmospheric Remote Sensing, Wuhan430072, China

*Correspondence to*: Fan Yi (yf@whu.edu.cn)

**Abstract**

Mid-level stratiform precipitations during the passage of warm front were detailedly observed on two occasions (light and moderate rain) by a 355-nm polarization lidar and water-vapor Raman lidar, both equipped with waterproof transparent roof windows. The hours-long precipitation streaks shown in the lidar signal ($X$) and volume depolarization ratio ($\delta_v$) reveal some ubiquitous features of the microphysical process of precipitating hydrometeors. We find that for the light rain case, precipitation that reaches the surface begins as ice-phase-dominant hydrometeors fall out of a shallow liquid cloud layer at

altitudes above the 0 °C isotherm level, and the depolarization ratio magnitude of falling hydrometeors increases from the liquid-water values ($\delta_v< 0.09$) to the ice/snow values ($\delta_v> 0.20$) during the first 100–200 m of their descent. Subsequently, the falling hydrometeors yield a dense layer with an ice/snow bright band occurring above and a liquid-water bright band occurring below (separated by a lidar dark band) as a result of crossing the 0 °C level. The ice/snow bright band might be a manifestation of local hydrometeor accumulation. Most falling raindrops shrink or vanish in the liquid-water bright band due

to evaporation, whereas a few large raindrops fall out of the layer. We also find that a prominent $\delta_v$ peak (0.10–0.40) always occurs at an altitude of approximately 0.6 km when precipitation reaches the surface, reflecting the collision-coalescence growth of falling large raindrops and their subsequent spontaneous breakup. The microphysical process (at ice-bright-band altitudes and below) of moderate rain resembles that of the light rain case, but more large-sized hydrometeors are involved.

## 1 Introduction

An observation-based understanding of the microphysical processes of precipitation is essential for weather/climate modeling and predictions. Such processes are difficult to observe since they involve a variety of hydrometeor sizes, shapes and phases at different altitudes, all of which are affected by cloud dynamics (Aggarwal et al., 2016; Pfitzenmaier et al., 2018). In situ aircraft observations deliver data on the sizes and numbers of hydrometeors only for small sampling volumes at single altitudes at any given time during preplanned case studies (Barrett et al., 2019). Although lidar and radar can measure the time-resolved vertical profiles of bulk backscattering quantities, retrievals of the microphysical properties of hydrometeors requires numerous assumptions (e.g., the hydrometeor shape and size distributions). Furthermore, in most cases, ground-based lidar cannot penetrate high enough to sample complete precipitating hydrometeor layers (only profiling the lower part of a layer) due to signal attenuation. Thus information about their source clouds is usually not available (Sassen et al., 2005; Di Girolamo et al., 2012; Mega et al., 2012). There is also a lack of systematically observed lidar data on precipitation processes because most lidar systems are not protected from precipitation. Cloud/precipitation radars are insensitive to small raindrops and droplets in cloud layers. Therefore, the microphysical processes of precipitation formation are not well understood thus far.

Satellite observations have revealed that cold clouds are the major source of liquid precipitation over land (Mülmenstädt et al., 2015). Heterogeneous ice formation pertinent to cold clouds is believed to lead to the generation of rain (Field and Heymsfield 2015; Bühl et al., 2016; Pfitzenmaier et al., 2018). The ice formation process has been studied extensively by observing liquid-layer-topped ice virgae because ground-based lidar and radar can reliably sample the entire height ranges of ice virgae and their apparent source cloud bases (Ansmann et al., 2009; de Boer et al., 2011; Bühl et al., 2016; Bühl et al., 2019).In stratiform cloud layers at temperatures above −20 °C, precipitating bulk ice particles (ice virga) occurred after bulk liquid phases had formed overhead (Ansmann et al., 2009; deBoer et al., 2011). This suggests that the heterogeneous nucleation of ice proceed via the freezing of supercooled droplets (Ansmann et al., 2009; deBoer et al., 2011). Our polarization lidar observations have revealed the detailed vertical structures of falling virgae and their supercooled liquid source cloud layers, indicating that the depolarization ratio values of falling hydrometeors increase rapidly with decreasing altitude on the top of the virgae (Cheng and Yi, 2020).

To study the microphysical processes that occur at altitudes ranging from the apparent source cloud base down to the near-surface during surface precipitation, a 355-nm polarization lidar and a water vapor Raman lidar at the Wuhan University atmospheric observation site were equipped with waterproof transparent roof windows. According to an artificial water splashing experiment, water accumulation on the lidar roof windows yielded nearly height-independent lidar signal ($X$,

range-corrected signal) attenuation, whereas neither the $X$ vertical structure nor the profile of the volume depolarization ratio $\delta_v$ (the magnitude and vertical structure) were altered. In addition, water accumulation on the roof windows hardly impacted the lidar-observed subcloud water vapor mixing ratio ($q_v$) profiles. This allows us to systematically observe precipitation processes (light and moderate rains). Based on our lidar observations obtained on two warm front occasions, a complete microphysical process is revealed for precipitating hydrometeors pertinent to warm-front-related mid-level stratiform precipitation (the ice nucleating processes are not covered). This paper first depicts the relevant instrumentation and methodology. Section 3 presents two light and moderate warm-front precipitation cases observed at our lidar site. The summary and conclusions are given in section 4.

## 2 Instrumentation and Methodology

### 2.1 Lidar

Precipitating hydrometeor observations were obtained with two newly developed lidars equipped with waterproof transparent roof windows at the Wuhan University atmospheric observatory (30.5°N, 114.4°E, 73 m above sea level). The roof windows were designed to project out from the surroundings, avoiding a heavy water accumulation on the window glass during rainfall. The two lidars can simultaneously deliver the sequential profiles of the range-corrected signal $X$, volume depolarization ratio $\delta_v$ and water vapor mixing ratio $q_v$. All the observation sessions started with clear-sky conditions and ended when heavy precipitation occurred. This allowed us to capture the evolving layer structures of light and moderate precipitation events as well as their precursor clouds present over our mid-latitude site.

### 2.1.1 Polarization Lidar

The polarization lidar has a configuration similar to our 532-nm system (Kong and Yi, 2015), but the transmitter employs a frequency-tripled Nd:YAG laser. It produces emissions of ~150 mJ per pulse at 355 nm with a repetition rate of 30 Hz. A Brewster polarizer is added to improve the polarization purity of the transmitting laser (up to ~10000:1). After beam expansion, the beam with a divergence of 0.15 mrad is transmitted vertically into the atmosphere (to the zenith). The backscattered light is collected by a 20-cm Cassegrain telescope. The field of view (FOV) of the receiver is ~1 mrad. After collimation, the elastically backscattered light passes an interference filter (with a 0.3-nm bandwidth centered at 355 nm) and then incidents a polarization beam splitter prism (PBS). To decrease the cross talk between the two orthogonal polarization channels, two additional polarizers are placed on the two output sides of the PBS. The light exiting from the two polarizers is focused onto two photomultiplier tubes (PMTs). The signals from the two PMTs are gathered by a PC-controlled two-channel transient digitizer (TR40–160, manufactured by Licel).

The raw lidar data are stored in both analog and photon counting modes with a range resolution of 3.75 m and a temporal resolution of 1 min. Based on a method originally proposed by Newsom et al. (2009) that was further developed by Zhang et al. (2014), the stored analog and photon-count data are glued to form a reasonable photon-count profile with a large dynamic range. For the cases in this study, the altitude range of signal gluing was ~1.2–3.3 km. The range and temporal resolution of the processed photon count profiles are 30 m and 1 min, respectively. The starting altitude of the lidar measurements is ~0.3 km, determined based on the overlap of the laser and the field of view of the telescope. The altitude values referenced in this article are all relative to sea level.

The range-corrected lidar signal $X$ is utilized to represent the backscattering intensity (returned laser power) of cloud particles and gravitationally-falling hydrometeors (Ansmann et al., 2008). The volume depolarization ratio $\delta_v$, defined by the ratio of the perpendicular- to parallel-polarized backscatter coefficients, can be obtained from the two-channel lidar signals along with the relative gain of the parallel and perpendicular channels. The relative gain is determined in advance using a conventional method (Freudenthaler et al, 2009). The magnitude of the $\delta_v$ value allows us to identify whether the dominant backscattering is attributed to ice crystals or water droplets in a given backscatter volume (Shupe, 2007). In general, liquid water droplets suspended in the atmosphere are nearly spherical and produce a very low depolarization ratio (close to zero) for single scattering at exact 180°, while ice crystals, which are usually nonspherical, generate a quite large depolarization ratio in the 180° backscattering direction. For some mid-level stratiform precipitations, gravitationally-falling hydrometeors form initially at altitudes above the 0 °C isotherm level. They fall often as ice-phase-dominant hydrometeors at sub-zero temperatures during their early descent. After the falling hydrometeors pass through the 0 °C isotherm level, the snowflake (ice)-to-raindrop transition can yield a shallow layer of relatively smaller lidar echoes (a local $X$ minimum), that is called "lidar dark band" (Sassen and Chen, 1995; Di Girolamo et al., 2012). The lidar dark band can be used to differentiate between the altitudinal regions with ice-containing particles above the dark band and pure liquid raindrops below the dark band.

It should be mentioned that the particle depolarization ratio $\delta_p$ is conceptually a more suitable quantity in discriminating spherical and nonspherical particles (hydrometeors) in virga/cloud than the volume depolarization ratio $\delta_v$. But, the volume depolarization ratio $\delta_v$ represents the more basic lidar measurement. In order to validly utilize the $\delta_v$ magnitude in discriminating spherical and nonspherical depolarizations, we have examined the relationship between $\delta_p$ and $\delta_v$. The $\delta_p$ magnitude is a well-defined function of $\delta_v$, lidar backscattering ratio $R$ and molecular depolarization ratio $\delta_m$ (Cairo et al., 1999). The molecular depolarization ratio $\delta_m$ has a value of ~0.004 in terms of our lidar receiver bandwidth (0.3 nm) (Behrendt and Nakamura, 2002). Information about the $R$ value range is available from a combined consideration of the earlier lidar measurements and our current observations on precipitation-related cloud/virga. The typical values of $R$ for enhanced aerosol load are around 2, for optically thin clouds up to around 10 (Lampert et al., 2010). The $R$ values are ~5–8

on the upper part of typical shallow (~400-m thick) evaporating ice virgae (see Figure 4 in (Cheng and Yi, 2020)). In this study, the $R$ value should certainly be larger than 7 on the precipitation-related virga layer. Based on the analysis to the $\delta_p$ expression (Cairo et al., 1999) for clouds and virgae, the particle depolarization ratio $\delta_p$ has a quasilinear dependence on the volume depolarization ratio $\delta_v$, and a very weak dependence on lidar backscatter ratio $R$ (when $R \geq 5$). This favorable feature of the functional dependences allows us to utilize $\delta_v$ in discriminating whether the dominant lidar backscattering is attributed to spherical or nonspherical particles in a given backscatter volume. If $R_{min}$ is the minimum of the $R$ value range for the interested clouds/virgae (e.g., $R_{min} = 7$ for the precipitation-related virgae in this study), the discrimination criterion of spherical particles expressed by $\delta_v(\mathrm{z})$ (equivalent to $\delta_p < 0.1$) takes the form (see Appendix A for mathematical derivation)

$$\delta_v(\mathrm{z}) < 0.1 - \frac{0.11}{R_{min} + 0.1} \ . \qquad (1)$$

The discrimination criterion of nonspherical particles expressed by $\delta_v(\mathrm{z})$ (equivalent to $\delta_p > 0.2$) is given approximately by

$$\delta_v(\mathrm{z}) > 0.2 - \frac{0.24}{R_{min} + 0.2} \ . \qquad (2)$$

As noticed from the right-hand sides of Inequalities (1) and (2), the absolute differences between the discrimination threshold values expressed by $\delta_p$ (0.1 and 0.2) and by $\delta_v$ are small for clouds/virgae with $R_{min} > 7$. The unambiguous cloud-phase discriminations based on the volume depolarization ratio $\delta_v$ in earlier literatures (Wang and Sassen, 2001; Intrieri et al., 2002; Shupe, 2007; Ansmann et al., 2009; Lampert et al., 2010) have confirmed the functional relationship between $\delta_p$ and $\delta_v$ mentioned above. This allows us to employ $\delta_v$ with very little threshold-value change in discriminating whether the dominant lidar backscattering is attributed to spherical or nonspherical particles in a given backscatter volume. Specifically, at altitudes above the dark band, the $\delta_v$-based discrimination criteria are $\delta_v < 0.09$ for spherical water drops/droplets and $\delta_v > 0.17$ for ice crystals (based on the above discrimination criteria when $R_{min} = 7$), while an enhanced depolarization ratio ($\delta_v > 0.1$) at altitudes below the dark band indicates the presence of large raindrops.

We examined the multiple-scattering-induced depolarization ratio enhancements for an opaque cloud layer composed of dense spherical water droplets by putting a motorized iris on our polarization lidar system. It is indicated that for a receiver FOV of 1 mrad, the enhanced depolarization ratio $\delta_v$ values due to multiple scattering increased from ~0.03 at the $X$ peak altitude to a maximum value of ~0.27 at the weak-signal cutoff altitude with increasing penetration of laser light into the opaque water-droplet cloud layer. Note that for the same receiver FOV (~1 mrad), the multiple-scattering-induced

depolarization ratio $\delta_v$ values were all less than 0.04 within the laser light penetration range in a slightly-dense water-droplet cloud layer (Hu et al., 2006). Combining the earlier multiple-FOV polarization lidar measurements (Hu et al., 2006) and our similar observations yields a suggestion that for the 1-mrad receiver FOV, the multiple-scattering-induced depolarization ratio values larger than 0.10 should result from an opaque water-droplet cloud layer (see Figs. 2 and 4 in (Yi et al., 2021)). In other words, for the 1-mrad receiver FOV, the vertical structure of hydrometeors and aerosols present above a dense water-

droplet cloud layer with $\delta_v$ values larger than 0.1 is undetectable by ground-based lidars. An artificial water splashing experiment was performed on the lidar roof windows to examine the effects of water accumulation. A comparison of the lidar profiles with and without water accumulation on the lidar roof windows is given in Figure 1. Enhanced lidar signal ($X$) and depolarization ($\delta_v$) values at altitudes around 4.0 km resulted from an optically-thick (opaque)water-droplet cloud layer because there existed a high $X$ value and near-zero $\delta_v$ value (~0.008) on the cloud base (~3.9 km) (Wang and Sassen, 2001),

and also there initially existed a monotonic rapid increase in both the values of $X$ and $\delta_v$ with increasing penetration of laser light into the layer. The cloud-related structures shown in both the $X$ and $\delta_v$ profiles were consistent before and after water splashing (particularly, cloud base altitudes). This comparison clearly shows that water accumulation on the lidar roof windows yielded nearly height-independent lidar signal ($X$) attenuation, and neither the cloud-related $X$ vertical structure nor the profile of the volume depolarization ratio $\delta_v$ (the magnitude and vertical structure) were altered. This result is physically

reasonable.

### 2.1.2 Water Vapor Raman Lidar

The configuration of the water vapor Raman lidar used in this study is similar to our 45-cm-aperture Raman system (Wu and Yi, 2017), but the current Raman lidar shares the same transmitter with our 355-nm polarization lidar depicted above. It

detects inelastic Raman backscatter from water vapor at 407 nm and nitrogen molecules at 387 nm as well as detecting elastic backscattered light by using a 20-cm receiver telescope. The water vapor mixing ratio $q_v$, which is defined as the mass ratio between water vapor and dry air in a given volume, can be obtained from the Raman signals representing water vapor and nitrogen molecules (Whiteman et al., 1992). The Raman lidar system was calibrated by corresponding local radiosonde measurements. A comparison analysis showed that the lidar-derived water vapor mixing ratio profiles agree well with the

coincident radiosonde data (the relative deviation is less than 10% when the water vapor field is horizontally homogeneous on a scale of ~20 km). During the daytime, the water vapor Raman signal is quite noisy at high altitudes due to strong sky background light, so the water vapor mixing ratio profiles are available only at altitudes below ~2 km. A similar artificial water-splashing experiment to that described above was performed on the water vapor Raman lidar roof window. Water accumulation on the roof window hardly has an impact on the obtained subcloud $q_v$ profiles.

## 2.2 Radiosonde

The radiosondes are launched at 0800 LT (0000 UTC) and 2000 LT (1200 UTC) every day from the Wuhan weather station (~23.4 km away from our lidar site). Profiles of the air pressure, temperature, relative humidity, wind speed, and direction from the near surface up to a height of 20–30 km are measured. The obtained radiosonde profiles are used to quantitatively determine the meteorological conditions pertinent to the precipitation events and their precursor clouds. The temperature measurement error is less than 1°C, and the uncertainty in the relative humidity measurement is less than 5% when the temperature is higher than 10°C (Nash et al., 2011).

## 2.3 All-sky camera and rain gauge

The cloud photographs are recorded every two minutes by a ground-based all-sky camera located at our lidar site. A tipping-bucket rain gauge is used to measure the precipitation rate at the surface. It has a sampling interval of 1 min. For each 0.1 mm of precipitation, the bucket tips and empties, yielding an output signal.

## 3 Observational Results

### 3.1 Light warm-front precipitation (26-28 December 2017)

Figure 2 presents an example of lidar observations obtained during a warm front passage and the resulting light precipitation. As seen from Figs.2a and 2b, a varying cloud layer descended steadily from ~10.3 km at ~1600 LT on 26 December to ~3.0 km at approximately 2351 LT on 27 December 2017. The cloud layer was mostly characterized by a mixed phase and had subcloud ice virgae during the later descent (Figs.2a and 2b). After the subcloud virgae reached an altitude (~3.0 km) that was lower than the 0°C level (~3.6 km), as measured by a conventional radiosonde at approximately 2000 LT on 27 December at the Wuhan weather station (~23 km away from our lidar site), falling raindrops (precipitation streaks in the $X$ and $\delta_v$ contour plots) that reached the ground were frequently observed beneath the 3-km altitude until 0538 LT on 28 December 2017 when the lidar operation terminated. Long survival time of falling ice crystals at altitudes below the 0°C level might be ascribed to cooling of the surrounding air during their evaporation and melting. The associated water vapor mixing ratio, $q_v$, increased steadily with the descent of the cloud layers (Fig.2c). In particular, a high-concentration moisture layer appeared in the subcloud region during the rainfall event. This moisture layer resulted mainly from the evaporation of snow/ice particles and raindrops. Corresponding photographs of the sky taken by a ground-based camera at our lidar site are given at the top of Fig.2. The light rain lasted for ~8 h and yielded an accumulated rainfall amount of 2.6 mm (rain gauge data obtained at our lidar site). Interestingly, a humid aerosol layer also moved downward from ~4.2 km at ~2000 LT on 25 December to ~2.3 km at 2000 LT on 27 December 2017, which appeared to be associated with the warm front.

### 3.1.1 Associated meteorological conditions

Figure 3 presents the radiosonde profiles that are pertinent to the warm-front cloud at different stages and during precipitation, together with the 1-h mean lidar profiles obtained during the radiosonde launches. The temporally-varying cloud properties (e.g., falling cloud base, increasing cloud thickness, and variable cloud types) between 2000 LT on 26 December and 2000 LT on 27 December 2017 coincided with the classical picture of preceding upglide clouds of an advancing warm-front system. Accordingly, a downgoing moist layer was observed strengthening and broadening with time during this period (Figs. 3b and 3c). At the cloud base (except cirrus), the relative humidity over ice had values close to the relative humidity threshold of 84% that is conventionally used to determine the cloud base heights (Wang and Rossow, 1995; Zhang et al., 2018). Furthermore, the radiosonde data exhibited that the southwesterly wind mostly prevailed at the cloud altitudes (Figs. 3d, 3e, and 3f), and the air pressure at altitudes of ~0−5 km dropped continuously by ~3−5 hPa in the period (not shown here), which did belong to the typical warm-front features.

The radiosonde released at 0800 LT on 28 December 2017 provided measurements of the meteorological conditions when precipitation reached the surface, although the lidar measurements had already terminated (at 0538 LT) ~ 2 hours earlier. As seen from Fig. 3b (red), the relative humidity reached a maximum of 98% with respect to water in an altitude range of ~3−4 km, immediately above the tops of the liquid precipitation streaks (at ~3 km, see Figs. 2a and 2b). Water vapor at altitudes of ~3−9 km was advected from the southwest, as seen in the wind component profiles (Fig. 3f, red). The high water vapor mixing ratios observed at altitudes below ~3 km came from the evaporation of falling raindrops.

### 3.1.2 Microphysical process of precipitating hydrometeors for the light warm-front rain

The $X$ and $\delta_v$ precipitation streaks were visible in the period between 2351 LT on 27 December and 0536 LT on 28 December2017 (Figs. 2a and 2b). The streaks extended from the starting height (~0.3 km) of the lidar measurements to an altitude of ~ 2.88 km when surface precipitation occurred. A lidar dark band ($X$ minimum) appeared persistently on the top of the precipitation streaks at an ~2.88-km altitude except when the dark band was concealed by a drifting small-scale cloud (at 2.2−2.6-km altitudes during 0418−0536 LT on 28 December). This is consistent with earlier lidar observations of stratiform precipitation (Sassen and Chen, 1995; Demoz et al., 2000; Roy and Bissonnette, 2001; Di Girolamo et al., 2012). An inapparent local depolarization ($\delta_v$) minimum was also persistently present at an altitude of~2.76 km, lying just ~100 m below the dark-band minimum (Fig.2b). The local $\delta_v$ minimum represented the completion of the melting process of most falling ice/snow particles. Note that the $\delta_v$ value decreased as a whole from the ice/snow (including partially melted large particles) values (0.10−0.34) at altitudes above the lidar dark band to the small liquid drop level ($\leq 0.04$, far less than the $\delta_v$-based discrimination threshold value of spherical particles when the lidar backscatter ratio $R \geq 5$) at an altitude ~100m below the dark-band minimum. The lidar dark band definitely differentiates the altitude regions of precipitating ice-containing

hydrometeors occurring above and liquid raindrops occurring below. Although the rainfall-induced water accumulation on the roof window of the lidar varied with time, the precipitation streaks and dark band were steadily reasonably displayed in the $X$ and $\delta_v$ time-height plots (Figs.2a and 2b). This is consistent with the result of our water splashing experiment.

To further clarify the microphysical process of precipitating hydrometeors, two sets of representative lidar profiles ($X$, $\delta_v$ and $q_v$) for the period that precipitation reached the surface (in Fig.2) are plotted in Figs. 4 and 5. Figure 4 gives three 1-min $X$ and $\delta_v$ profiles from 0112 to 0114 LT on 28 December 2017 and a one-hour-averaged $q_v$ profile centered at 0113 LT on the same day. The lidar dark band appeared at a 2.88-km altitude at approximately 0113 LT, while the local $\delta_v$ minimum (< 0.04, far less than the $\delta_v$-based discrimination threshold value of spherical particles when $R \geq 5$) was located at a 2.76-km altitude. These altitudes represent a typical lidar signature of the snowflake-to-raindrop transition for a variety of stratiform precipitation events. An ice-containing bright band (ice bright band hereafter) with $\delta_v$ values ranging from ~0.13 to ~0.39 was visible at altitudes ~3.0–3.45 km, just above the lidar dark band (Fig. 4); these altitudes correspond to the "relative lidar bright band" in the literatures (Sassen and Chen, 1995; Di Girolamo et al., 2012). The ice bright band peaked on its bottom (~3.0 km). It showed a variable vertical structure and intensity (in both $X$ and $\delta_v$) on the time scale of minutes, representing the presence of small-scale fluctuations in the precipitating ice crystals and snowflakes. A liquid water bright band appeared as a layer of relatively large particle backscatter values, located at ~1.50–2.76-km altitudes, just below the lidar dark band (Fig. 4). It is called "weak lidar bright band" in the literatures (Sassen and Chen, 1995; Di Girolamo et al., 2012). The $\delta_v$ values in the water bright band were ~0.03–0.06, indicating that the enhanced lidar backscattering therein was caused mainly by high-concentration quasi-spherical raindrops with diameters ≤ 1 mm. The water bright band actually represents a major precipitation-related lidar backscattering layer in the liquid-phase stage of the light precipitation event. The water bright band appeared to have a larger vertical extent (~1.26km) than that of the lidar ice bright band.

The lidar $q_v$ profile (Fig. 4c) shows an enhanced water vapor mixing ratio at altitudes from~1.7–3.4 km, indicating the subcloud evaporation of precipitating hydrometeors. In particular, $q_v$ was maximized (5.95 g kg$^{-1}$) around the water bright band center (at ~2.34 km), suggesting that this altitude was a primary subcloud evaporation region for this light warm-front precipitation event. Furthermore, the $q_v$ values in the water bright band increased as precipitation continued (Fig.2c). Combining the vertical structures of $X$, $\delta_v$, and $q_v$ in the water bright band (Figs. 2 and 4) yields the suggestion that most falling small-sized raindrops shrunk or vanished in the water bright band due to evaporation, whereas a small portion of large-sized raindrops survived via collision-coalescence processes and fell out of the water bright band.

At altitudes below the water bright band, the precipitation-related lidar backscattering ($X$) apparently weakened (Fig. 4a, in which the enhanced $X$ values at altitudes from 0.3–0.7 km resulted from the boundary layer aerosols), indicating low-density raindrops there, whereas $\delta_v$ first increased with decreasing height and then decreased after reaching a maximum (0.13–0.16)

at an altitude of approximately 0.6 km (Fig. 4b). Here the magnitude and altitude variation of the lidar depolarization ratio $\delta_v$ values allow us to identify where large-sized raindrops form and break up. Falling small-sized raindrops (equivalent diameter ≤ 1.0 mm) are quasi-spherical (Pruppacherand Klett, 1997) and yield small $\delta_v$ values (generally less than 0.1),

whereas falling large-sized raindrops (equivalent diameter > 2.8 mm) become nonspherical (with flat or hollow bottom in falling direction) (Pruppacherand Klett, 1997) and lead to large $\delta_v$ values (larger than 0.1). In fact, prominent $\delta_v$ peaks (~0.1–0.4) at altitudes of approximately 0.6 km are always observed in the $\delta_v$ profiles related to reaching-surface precipitation in the present light rain case (Fig. 2). The $\delta_v$ maxima at an altitude of ~0.6 km are much larger than the typical values (<~0.07) observed by our 355-nm polarization lidar at approximately the same altitude during rainless days. Here we

can exclude a possibility that the $\delta_v$ maxima (~0.1–0.4) at ~0.6-km altitude resulted from multiple scattering by dense droplets around this altitude. As mentioned above, for the 1-mrad receiver FOV, a dense water-droplet cloud layer with the multiple-scattering-induced depolarization ratio $\delta_v$ values larger than 0.1 is optically opaque. In contrast to this situation, in our case, when the prominent $\delta_v$ peak (~0.1–0.4) around 0.6-km altitude occurred, the vertical structure of the precipitation streaks at altitudes far above 0.6 km (e.g., ice bright band, lidar dark band and lidar water bright band) was unambiguously

detected by our polarization lidar, indicating that the enhanced depolarization ratios around 0.6-km altitude cannot be caused by multiple scattering from dense spherical water droplets therein. Furthermore, since most falling raindrops evaporated and vanished in the liquid-water bright band as indicated by the enhanced water vapor mixing ratio therein and rapidly-decreasing lidar signal on the bottom of the water bright band, small droplets at altitudes below the water bright band were hardly dense enough to generate a strong multiple scattering with $\delta_v$≥ 0.1.Therefore,our observational results suggest that

sparse large raindrops that fall out of the water bright band with higher fall velocities further grow in size by collecting smaller raindrops along their fall paths. They grow to sizes at which spontaneous breakup occurs at an altitude of approximately 0.6 km. In brief, our lidar observations reveal for the first time (to our knowledge) the collision-coalescence growth and subsequent spontaneous breakup of falling raindrops, that actually take place in the natural atmosphere. They represent the posterior microphysical processes necessary for the reaching-surface precipitation production. Interestingly, the

size maximization of falling raindrops as shown by the strongest nonspherical shapes (maximum depolarization ratio values) always appeared at an altitude of ~0.6 km for a variety of mid-level stratiform precipitations (in light of our observations). Obviously, the explanation to this ubiquitous feature needs further observational and modelling efforts. As seen in Fig. 2b, the boundary layer aerosols had little impact on the $\delta_v$ precipitation streaks. In addition, at altitudes below 1.5 km, the $q_v$ values decreased with increasing altitude, reflecting a normal altitude distribution of the boundary layer water vapor.


Based on the radiosonde temperature data obtained at approximately 2000 LT on 27 December 2017 (Fig.3a, orange), the 0°C isotherm level was at an altitude of ~3.6 km, and a warm-front-related inversion layer appeared just below the 0°C level with a local temperature maximum (2.2 °C) at 3.33 km and a local minimum (1.0 °C) at 2.84 km. The lidar dark band (at 2.88 km, with a temperature of ~1.0 °C) was located ~720 m below the 0°C level. In comparison with the results reported in

the literatures (Sassen and Chen, 1995; Demoz et al., 2000; Sassen et al., 2005; Di Girolamo et al., 2012), the observed~720-m distance of the dark-band minimum to the 0°C level and the low dark-band temperature (~1.0 °C) are somewhat peculiar for light precipitation cases. In the current case, the melting process might be delayed by the temperature structure (with a small lapse rate) of the inversion layer. However, it should be mentioned here that the radiosonde launching site was ~23.4 km away from our lidar site.


Figure 5 presents three 1-min lidar $X$ and $\delta_v$ profiles displaying the timespan from 0230 to 0232 LT on 28 December 2017 and a one-hour-averaged lidar $q_v$ profile centered at 0231 LT on the same day, depicting the microphysical process of precipitating hydrometeors for slightly strong precipitation that reached the surface during the light rain event. Although the water bright band and aerosol backscatter layer below the dark band became evidently weak compared to those seen in

Fig.4a (due to precipitation attenuation), the altitude of the dark-band minimum (2.85 km) was very close to that (2.88 km) obtained from Fig.4a. The magnitude (~0.03) and occurrence altitude (2.76 km) of the local $\delta_v$ minimum were consistent with the corresponding values (less than 0.04 and 2.76 km, respectively) observed in Fig. 4b. Furthermore, the depolarization maxima (~0.17) associated with reaching-surface precipitation still appeared at an altitude of approximately 0.6 km, which was also similar to that seen in Fig.4. The observational facts confirm the result gathered from our water splashing

experiment in which the thin liquid water layer on the roof windows of the lidars caused nearly altitude-independent attenuation on the $X$ profiles and had no effect on the $\delta_v$ profiles. The profile characteristics shown in Fig.5 are mostly similar to those mentioned above for Fig.4, but some newly emerging features need to be illustrated. Figure 5 exhibits an ice bright band stronger than the concurrent water bright band. This result is different from our observations obtained at ~0112 LT (Fig.4) but is consistent with earlier lidar observations (Sassen and Chen, 1995; Di Girolamo et al., 2012). The ice bright

band observed at approximately 0230 LT had the $X$ maxima at its bottom (at an altitude of~3.0 km) and a small vertical extent (~0.21 km, from 3.00 to ~3.21 km due to precipitation attenuation). The $X$ maxima corresponded to the local minima of the depolarization ratio (Figs. 5a and 5b). Interestingly, this inverse relationship between the backscatter and depolarization values on the bottom of the ice bright band is nearly ubiquitous in the precipitation lidar profiles obtained in the present case. Since the depolarization $\delta_v$ showed moderate minima (~0.08–0.10) at an altitude of ~3.0 km (Fig.5b), the

ice bright band maxima observed at approximately 0230 LT might reflect backscattering from high-concentration partially melted large particles therein. On the band's altitudinal extension (from 3.06–3.21 km), the markedly enhanced depolarization values (~0.17–0.34) indicate the presence of ice crystals and large snowflakes (Sassen and Chen, 1995; Di Girolamo et al., 2012). The water vapor mixing ratio $q_v$ showed slight enhancements at altitudes of ~1.5–3.0 km at approximately 0230 LT compared with that measured at approximately 0112 LT.


As seen from the $X$ and $\delta_v$ precipitation streaks at altitudes below ~1.5 km (Figs.2a and 2b), precipitation that reached the surface was intermittent. During periods without reaching-surface precipitation, our lidars were able to sample both a

complete virga (from the rain to the snow regions) and a shallow mixed-phase cloud layer immediately above the virga under weak optical attenuation conditions. Such an example is shown in Fig.6. The lidar profiles above the dark band clearly

exhibit the typical structure characteristics of a liquid-topped mixed-phase cloud (a shallow liquid cloud layer and ice virga below) (see Fig.6 in (Wang and Sassen, 2001)). The mixed-phase cloud top layer (at altitudes of ~4.6 km) was of large $X$ values and very low $\delta_v$ values (~0.01), while lower part of the cloud was characterized by significantly-smaller $X$ values and large $\delta_v$ values (with a maximum up to ~0.33). Furthermore, the cloud top layer had a maximum water vapor mixing ratio $q_v$ and a temperature of ~−8.5 °C (based on radiosonde data at ~2000 LT on 27 December). Combining with the schematic

representation of commonly-observed mixed-phase cloud layers (see Fig. 1 in (Bühl et al., 2016)), the current observations suggest that the cloud top layer should mainly be composed of liquid droplets (that were not dense enough to yield detectable multiple scattering), and the lower part of the cloud was mainly precipitating ice crystals (falling ice virga). The liquid-topped mixed-phase cloud (a liquid cloud layer and ice virga below) (Bühl et al., 2016) might be fundamental monomers that constitute mid-level precipitating stratiform clouds. According to the expressions on the right-hand sides of Inequalities (1)

and (2), the $\delta_v$-based discrimination threshold values were respectively 0.09 for spherical particles and 0.17 for nonspherical particles when the lidar backscatter ratio $R$ had a value of 7 (the minimum of the $R$ value range) on the upper part of the precipitation-related virga (Lampert et al., 2010; Cheng and Yi, 2020). Thus, the $\delta_v$ magnitude of the falling virga increased from the liquid-water values of ~0.02−0.07 (< 0.09) at an altitude of 4.38 km to the ice/snow values of ~0.21−0.33 (> 0.17) at an altitude of 4.02 km. The falling ice crystals yielded a very weak ice bright band at an altitude of ~3.0 km, and then

melted into liquid drops at an altitude of ~2.76 km (the local $\delta_v$ minimum). During their further descent, the liquid drops fully vanished due to evaporation, leaving a lidar-detectable rain virga (water bright band) without reaching-surface precipitation. In contrast to the situation during precipitation that reached the surface, no clear-cut $\delta_v$ enhancement occurred at an altitude of approximately 0.6 km when there were only virgae suspended in air. Similar results were discerned for other lidar profiles shown in Fig. 2, in which a complete mixed-phase cloud layer could be detected.


During the light warm-front rain event, since the reaching-surface precipitations and virgae occurred alternately on a small time scale from a few minutes to tens of minutes and since their precipitation streaks had nearly the same dark-band structures (Figs.2a and 2b), both reaching-surface precipitation and virgae would come from the same source cloud (because a warm-front cloud system is generally widespread and slowly varying). Reaching-surface precipitation (drizzle) arose when

the precipitation rate was high below the shallow water-droplet-dominated cloud layer (apparent source cloud), while virgae without reaching-surface precipitation took place when the subcloud precipitation rate was slightly low. Therefore, the current lidar observations reveal the microphysical process of precipitating hydrometeors related to light warm-front rain. Both reaching-surface rainfall and virgae suspended in air began as ice-phase-dominant hydrometeors fell out of a liquid apparent source cloud layer at altitudes above the 0 °C isotherm level. The depolarization ratio magnitude of falling

hydrometeors increased from the liquid-water values ($\delta_v$< 0.09) to the ice/snow values ($\delta_v$> 0.20) during the first 100−200 m

of their descent. Subsequently, the falling hydrometeors yielded a dense layer with an ice/snow bright band occurring above and a liquid-water bright band occurring below (separated by a lidar dark band) as a result of crossing the 0°C level. In the ice/snow bright band, large particles would form via the cold rain processes (riming and aggregation) because the broad size distributions of the pristine hydrometeors falling out of the apparent source cloud base could lead to local accretion. The production efficiencies of large particles would depend on the magnitude of the rain rate below the apparent source cloud base and size distributions of the pristine falling hydrometeors. The local depolarization minimum ($\delta_v \leq 0.04$, far less than the $\delta_v$-based discrimination threshold value of spherical particles when $R \geq 5$) was persistently observed immediately beneath (~100 m below) the lidar dark-band minimum ($X$ minimum). This displayed that the completion of the melting process of most falling ice particles took place at altitudes (hundreds of meters) below the 0°C isotherm level. The liquid-water bright band (with a geometrical thickness of ~ 1 km) just below the lidar dark band was characterized by enhanced $X$ values and small $\delta_v$ values. There existed a high-concentration moisture (large $q_v$ values) in this bright band. These features indicate that the liquid-water bright band resulted from gravitationally-falling, dense evaporating liquid drops. In terms of the lidar-measured profiles during reaching-surface precipitation, at altitudes below the water bright band, the precipitation-related lidar backscattering apparently weakened, while $\delta_v$ first increased with decreasing altitude and then decreased after reaching a prominent maximum at an altitude of ~0.6 km. The lidar profiles for the virgae showed narrower and weaker water bright bands than those observed when precipitation reached the surface. Moreover, during virga occurrence, there was no perceptible depolarization enhancement at an altitude of~0.6 km. By combining the above-mentioned lidar observations, a picture on the microphysical processes of falling hydrometeors in liquid-phase stage emerged. After going through the dark band, most falling raindrops shrank or vanished in the water bright band due to evaporation, whereas a few large raindrops survived and fell out of the water bright band when the rain rate below the apparent source cloud base was high enough. The large raindrops might come from both the complete melting of large falling ice/snow particles and collision-coalescence formation in the dense water bright band. Sparse, large raindrops with high fall velocities further grew in size by collecting smaller raindrops along their fall paths. At an altitude of ~0.6 km, the large raindrops grew to the sizes at which spontaneous breakup could occur, yielding reaching-surface precipitation. When the rain rate below the apparent source cloud base was low, nearly none of the large raindrops fell out of the water bright band. Consequently, there were only virgae suspended on air (without reaching-surface precipitation).

**3.2 Moderate warm-front precipitation (4 March 2019)**

Figure 7 shows an example of moderate warm-front precipitation that occurred on 4 March 2019. Both the descending precursor clouds and the $X$ and $\delta_v$ precipitation streaks are generally similar to those seen in the first example (Fig.2). The precursor clouds are cirrus (photo I in Fig. 7), altostratus (photo II) and altocumulus (photo III). The reaching-surface precipitation started just after the subcloud ice virgae reached an altitude (~2.7 km) slightly lower than the 0 °C level (~3.0

km). The $\delta_v$ precipitation streaks show the upper portion (ice bright band) containing ice/snow particles (mostly $\delta_v > 0.3$) and the lower portion (water bright band and below) being composed of liquid drops ($\delta_v \leq \sim 0.12$ except for the $\delta_v$ maxima that occurred due to raindrop-size growth at an altitude of~0.6km). The $\delta_v$ values in both the ice bright band and water bright band (Fig. 7b) were generally larger than their counterparts in the light rain example (Fig.2b), indicating that more large ice/snow particles and raindrops were involved in the moderate precipitation than in the light precipitation event. Partially melted, large falling particles sometimes concealed the lidar dark band produced by the melting effect of most relatively small-sized particles in precipitating hydrometeors, making the band somewhat fuzzy (Fig.7a). Accordingly, the altitude of the local $\delta_v$ minimum (on the lidar dark band) became somewhat unsteady (Fig. 7b). The $\delta_v$ maxima at an altitude of approximately 0.6 km (Fig.7b) were apparently larger than those shown in Fig.2b, indicating that more breakup-size raindrops formed via collision-coalescence processes therein than in the light rain case. Specifically, the $\delta_v$ maxima at an altitude of~0.6km were as high as ~0.27–0.35 at ~2338 LT, which corresponded well to the large rainfall rate of 3.2 mm h$^{-1}$ measured from our rain gauge on the ground. The (apparent) source cloud for this moderate rain event was invisible by the lidars due to strong optical attenuation. Therefore, the following analysis was limited to the ice bright band and below. A strong southerly wind prevailed at altitudes of 0–12 km in light of the radiosonde data obtained at 2000 LT on 4 March 2019. A high-concentration moisture layer appeared in the subcloud region at altitudes from ~0.5 to ~3.0 km during the rainfall event, indicating the subcloud evaporation of precipitating hydrometeors. The moderate rainfall lasted for ~14 h, yielding an accumulated rainfall amount of 23.9 mm on the ground.

### 3.2.1 Associated meteorological conditions

The conventional radiosonde profiles associated with the moderate warm-front precipitation and its precursor clouds and the 1-h mean lidar profiles obtained during the radiosonde launches are plotted in Fig.8. At ~0800 LT on 4 March 2019, the sky was nearly cloudless (Fig.8d, blue), and high relative humidity occurred only at altitudes below 1.2 km (Fig.8b, blue), while the northwesterly wind prevailed at altitudes from1.7–10.5 km. This indicated that the warm front had not yet reached our lidar site. At ~2000 LT on 4 March, a moist layer occurred at altitudes ranging from ~4.8 to 8.0 km with increased relative humidity over water of 80-95% (Fig.8b, green). An evaporating ice virga was observed at altitudes from~3.6–4.6 km (Figs.8d and 8e, green), just below the moisture layer peak. The apparent source cloud of the virga was invisible by lidars due to strong optical attenuation. A potential occurrence region for the apparent source cloud ranged in altitude from 4.8–6.0 km, where the relative humidity was larger than 90% (Fig.8b, green). The southerly wind prevailed at altitude from 0–12 km (Fig.8f, green), indicating that the moisture layer and altocumulus (photo III in Fig. 7) were precursors of the warm-front precipitation event. The radiosonde profiles obtained at 0800 LT on 5 March 2019 showed the meteorological conditions during the moderate warm-front precipitation event after the lidar measurements had already terminated (at 0051 LT on 5 March). As shown in Fig.8b (orange), the relative humidity over water had values of 97–98% at altitudes from 0–5.65 km, corresponding to a precipitation rate of ~1.8 mm h$^{-1}$ (rain gauge record) at approximately 0800 LT on 5 March. The air

pressure from the radiosonde data at Wuhan showed a persistent decrease (by ~2–4 hPa at altitudes of ~0–5 km) during the observational period from the precursor clouds to precipitation (between 0800 LT on 4 March to 0800 LT on 5 March), that reflected the warm front passage.

### 3.2.2 Microphysical process of precipitating hydrometeors for the moderate warm-front rain

    Figure 9 presents three 1-min lidar $X$ and $\delta_v$ profiles measured from 2220 to 2222 LT on 4 March and a one-hour-averaged
lidar $q_v$ profile centered at 2221 LT on the same day; these profiles exhibit the vertical structure of the $X$ and $\delta_v$ precipitation streaks as well as the water vapor mixing ratio at the onset of the moderate warm-front precipitation event. The lidar profiles obtained at 2220 and 2221 LT show nearly identical dark-band locations (the $X$ minima is located at ~2.04 km, and the local $\delta_v$ minima is located at ~1.96 km). The dark-band minima appeared ~ 960 m below the 0°C level at a radiosonde temperature of ~6.0 °C. Such a long survival time of falling ice crystals at altitudes below the 0°C level was due to cooling of the
surrounding air during their evaporation and melting. At 2222 LT, a weak $X$ peak occurred at the dark-band altitudes with $\delta_v$ values ranging from~0.21–0.29, indicating that partially melted large particles passed through the dark band. As seen from Fig.9, the $X$ and $\delta_v$ precipitation streaks had complicated vertical structures at altitudes below the dark band and showed strong variations on the time scale of minutes. In particular, enhanced depolarization (0.07–0.12) occurred within the water bright band. These profile details confirm that large-sized particles sometimes fell out of the ice bright band during the
moderate warm-front precipitation event, concealing the lidar dark band produced by the melting effect of most relatively small-sized particles in precipitating hydrometeors. This effect appears to explain why the lidar dark band became fuzzy for the moderate precipitation event (Fig.7). Note that the $\delta_v$ maxima (~0.2), which occurred at an altitude of approximately 0.6 km, were slightly larger in the moderate warm-front rainfall than those observed in the light warm-front rainfall example. This suggests a larger concentration of raindrops of spontaneous breakup sizes around this altitude. The water vapor mixing
ratio $q_v$ had values ranging from 3.4–4.4 g kg$^{-1}$ at altitudes from 1.0–3.0 km (from the bottom of the water bright band to the 0°C level).

    Figure 10 gives three 1-min lidar $X$ and $\delta_v$ profiles representing the period from 2337 to 2339 LT on 4 March 2019 and a one-hour-averaged lidar $q_v$ profile centered at 2338 LT on the same day; these profiles exhibit the vertical structures of the
$X$and $\delta_v$ precipitation streaks as well as the water vapor mixing ratio observed when the surface precipitation rate was highest (3.2 mm h$^{-1}$) during the moderate warm-front precipitation event (yielding thick liquid water accumulation on the roof windows of the lidars; see photo V in Fig.7). The ice bright band, dark band and water bright band were roughly discernible in the three 1-min $X$ profiles despite the considerable fluctuations that occurred on the time scale of minutes. Large ice/snow particles occurred on the ice bright band (at an altitude of approximately 2.5 km) because the $\delta_v$ values were larger than 0.3
therein. The dark band located ~700 m below the 0 °C level (3.0 km) had $\delta_v$ values ranging from 0.13–0.19 and a temperature of 4.3 °C, reflecting that there were partially melted large particles present in the dark band. In the height range

of the water bright band, the depolarization ratio increased from ~0.04–0.06 at an altitude of approximately 2.09 km to ~0.12–0.15 at an altitude of 0.9 km, indicating that more large raindrops formed via collision-coalescence processes therein than in the light warm-front rainfall example (Figs.4 and 5). The $\delta_v$ maxima observed at an altitude of~0.6 km were as high as ~0.27–0.35 corresponding well to the high measured rainfall rate of 3.2 mm h$^{-1}$ (rain gauge record on the ground). As mentioned above, for the 1-mrad receiver FOV, if such large $\delta_v$ values (~0.27–0.35) came from the multiple scattering by a dense water-droplet cloud layer around 0.6-km altitude, the cloud layer would be optically opaque. It would conceal the vertical structure of the precipitation streaks at altitudes above 0.6 km. In contrast to this situation, as seen from Fig. 10, the vertical structure of the precipitation streaks at altitudes above 0.6 km was clearly discerned by our ground-based polarization lidar, indicating that the enhanced depolarization ratios around 0.6-km altitude cannot be caused by multiple scattering from dense spherical water droplets therein. Furthermore, since most falling raindrops evaporated and vanished in the liquid-water bright band as indicated by the enhanced water vapor mixing ratio therein and rapidly-decreasing lidar signal on the bottom of the water bright band, small droplets at altitudes below the water bright band were hardly dense enough to generate a strong multiple scattering with $\delta_v \geq 0.1$. Therefore, it is suggested that the prominent $\delta_v$ peak at an altitude of approximately 0.6 km reflected the collision-coalescence growth of falling large raindrops and their subsequent spontaneous breakup. The $q_v$ values at altitudes from~0.7–3.0 km ranged from 5.3–7.3 g kg$^{-1}$ (Fig.10c), indicating overall moisture enhancement compared to those values measured at the onset of the moderate warm-front precipitation event (Fig.9c).

## 4 Summary and conclusions

Observations of precipitation and associated precursor clouds were made with two co-located lidars (a 355-nm polarization lidar and water vapor Raman lidar) equipped with waterproof transparent roof windows at the Wuhan University atmospheric observatory (30.5°N, 114.4°E, 73 m above sea level).The lidar observations obtained during reaching-surface precipitation events indicate that the rainfall-induced liquid water accumulation on the roof windows of the lidars yielded a nearly height-independent lidar signal (range-corrected signal $X$) attenuation, whereas neither the $X$ vertical structure nor the magnitude or vertical structure of the volume depolarization ratio ($\delta_v$) were altered. Furthermore, the liquid water accumulation on the roof windows of the lidars also had nearly no effect on the obtained subcloud profiles of the water vapor mixing ratio measured by the Raman lidar. These observations are consistent with the results of our artificial water splashing experiment on the roof windows.

Warm-front precipitation events and their precursor cloud evolutions were reported in this paper based on two case studies corresponding to light and moderate rainfall occurring at the Earth's surface. The lidar-observed precursor clouds showed a systematic descent for each case. The descending clouds changed gradually from cirrus and altocumulus to altostratus before

rainfall occurred, with gradually increasing moisture, and the southwesterly wind prevailed over most altitude ranges of the cloud layers.These features indicate that, in each case, a warm front was approaching our lidar site. The precursor clouds had underlying ice virgae in their later descent phases. When the subcloud virgae reached an altitude slightly below the 0°C level, rainfall at the surface began. The hours-long precipitation streaks shown in the lidar signal ($X$) and volume depolarization ratio ($\delta_v$) profiles reveal some ubiquitous features of the microphysical processes of precipitating hydrometeors.

For the light warm-front rain event, since the reaching-surface precipitations and virgae occurred alternately over a short time scale from a few minutes to tens of minutes and since their respective precipitation streaks had nearly the same dark-band structures, both reaching-surface precipitations and virgae originate from the same source cloud (because a warm-front cloud system is generally widespread and slowly varying). Through an analysis combining the lidar profiles of reaching-surface precipitations and virgae, we find that the reaching-surface precipitation began as ice-phase-dominant hydrometeors fell out of a liquid apparent source cloud layer at altitudes above the 0 °C isotherm level. The depolarization ratio magnitude of falling hydrometeors increased from the liquid-water values ($\delta_v < 0.09$) to the ice/snow values ($\delta_v > 0.20$) during the first 100–200 m of their descent. Subsequently, the falling hydrometeors yielded a dense layer with an ice/snow bright band occurring above and a liquid-water bright band occurring below (separated by a lidar dark band) as a result of crossing the 0°C level. In the ice/snow bright band, larger particles formed by riming and/or aggregation because the broad size distributions of the pristine hydrometeors falling out of their apparent source cloud base could lead to local accretion. The completion of the melting process of most falling ice particles appeared at altitudes (hundreds of meters) below the 0°C isotherm level, as indicated by the local depolarization minimum located immediately beneath (~100 m) the observed lidar dark-band minimum. After going through the dark band, most falling raindrops shrunk or vanished in the water bright band due to evaporation, whereas a few large raindrops survived and fell out of the water bright band when the rainfall rate below the liquid apparent source cloud base was high enough. Large raindrops might originate from both the complete melting of falling large ice/snow particles and collision-coalescence formation in the dense water bright band. We also find that a prominent depolarization $\delta_v$ peak (0.10–0.40) always occurred at an altitude of approximately 0.6 km when precipitation reached the surface, reflecting the collision-coalescence growth of large falling raindrops (sparse large raindrops with high fall velocities further grew in size by collecting smaller raindrops along their fall paths) and subsequent spontaneous breakup. The $\delta_v$ peak observed at an altitude of ~0.6 km provides an indicator ~2mins in advance of precipitation that reached the surface.

For the moderate warm-front rain event, although the apparent source cloud was unable to detect owing to strong attenuation, the lidar-detectable microphysical process (at the altitudes of the ice-bright-band and below) was similar to that observed in the light rain case. However, the $\delta_v$ values in both the ice bright band and water bright band were generally larger than their counterparts in the light rainfall case, indicating that more large ice/snow particles and raindrops were involved in moderate

precipitation. Furthermore, the $X$ and $\delta_v$ precipitation streaks had complicated vertical structures at altitudes around and below the dark band and showed strong variations on the time scale of minutes. These profile details suggest that large particles sometimes fell out of the ice bright band during moderate precipitation, concealing the lidar dark band produced by the melting effect of most relatively small particles in precipitating hydrometeors. Thus, the lidar dark band became fuzzy.

The $\delta_v$ maxima observed at an altitude of approximately 0.6 km were also larger than those observed in the light warm-front rain case. This suggests larger concentrations of raindrops with spontaneous breakup sizes around this altitude.

**Author contributions**

YY performed the lidar measurements, made the data analysis and wrote the initial manuscript. FY conceived the project,
led the study and finalized the manuscript. FL, YZ and CY built the lidar systems for precipitation observations. YH participated in scientific discussions and suggested analysis. All authors discussed the results and commented on the manuscript.

**Competing interests**

The authors declare that they have no conflict of interest.

**Data availability**

Lidar data used in this work are available under permission (yf@whu.edu.cn).

**Acknowledgments**

This research is funded by the National Natural Science Foundation of China through grants 41927804, and also supported
by the Fundamental Research Funds for the Central Universities grant 2042021kf1006.The Meridian Space Weather Monitoring Project (China) also provides financial support for the lidar maintenance. The authors thank the University of Wyoming for providing the Wuhan radiosonde data at the website (http://weather.uwyo.edu/upperair/bufrraob.shtml). Data used to generate the results of this paper are available from the authors upon request (E-mail: yf@whu.edu.cn).

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

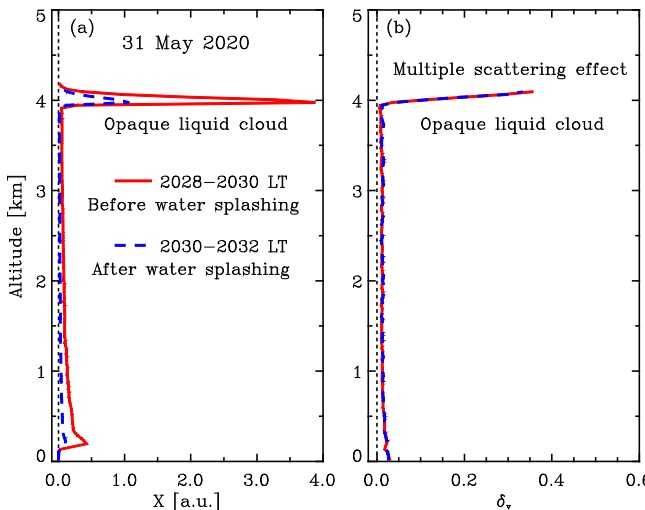

**Figure 1: Comparison of the lidar profiles with (integrated from 2030 to 2032 LT on 31 May 2020, dashed blue line) and without (integrated from 2028 to 2030 LT on the same day, solid red line) water accumulation on the lidar roof windows. (a), Range-corrected 355-nm signal $X$ profiles; (b), 355-nm volume depolarization ratio $\delta_v$ profiles. The water accumulation was produced by an artificial water splashing experiment.**


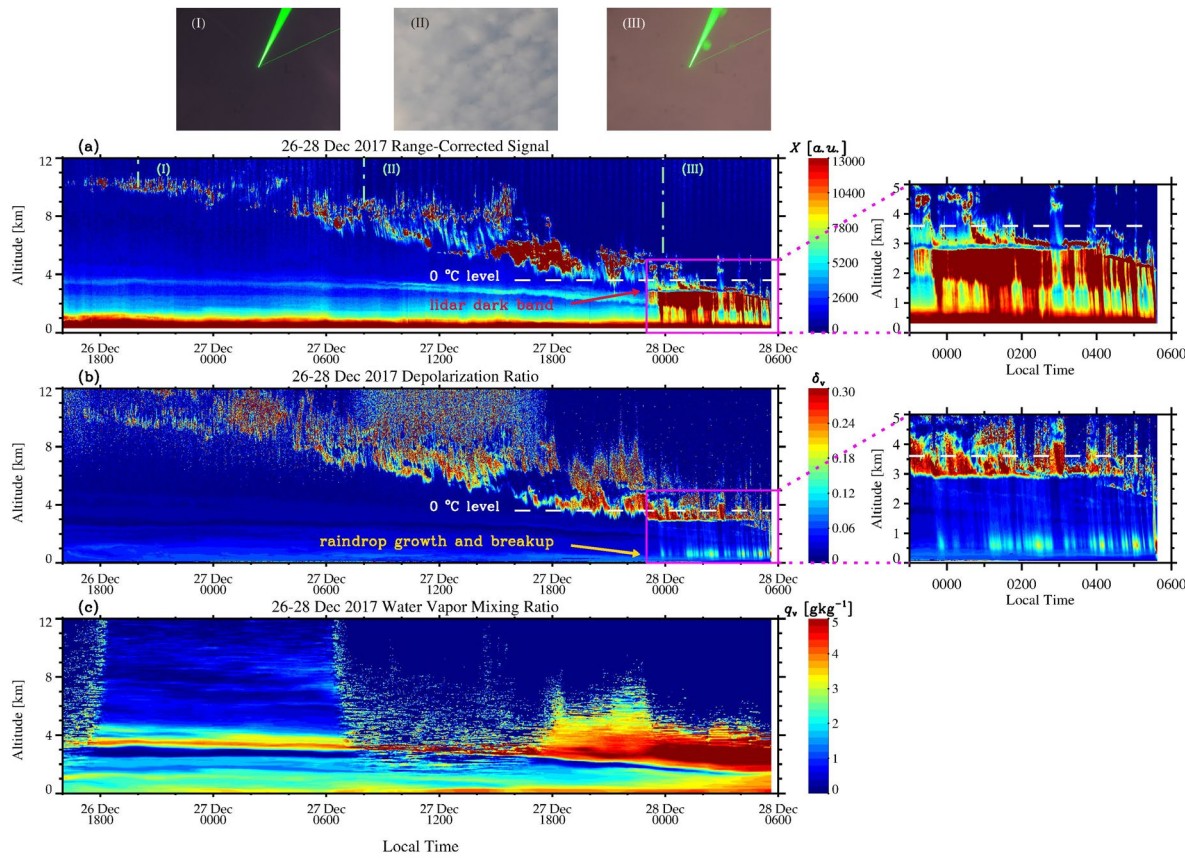


**Figure 2: Time-height contour plots (1 min/30 m resolution) of the (a) range-corrected signal *X*, (b) volume depolarization ratio $\delta_v$ measured by a 355-nm polarization lidar, and (c) water vapor mixing ratio $q_v$ measured by a water vapor Raman lidar on 26– 28December 2017, which exhibited the passage of a warm front and the resulting hours-long light rain. A sliding average of 60 min was applied to the Raman lidar data. The precipitation streaks surrounded by magenta lines are zoomed in to show their details.**
**Shown on the top of the figure are the corresponding photographs of the sky taken by a ground-based camera at our lidar site, with the third photograph exhibiting the sky illuminated by a 532-nm laser beam during the onset of rainfall.**

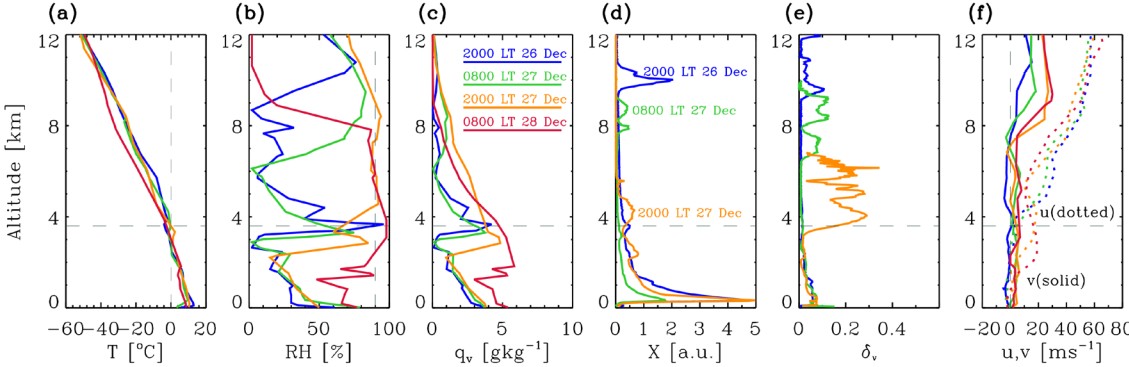

**Figure 3:** Sequential profiles of (a) temperature *T*, (b) relative humidity over liquid water *RH*, (c) water vapor mixing ratio $q_v$ and (f) the eastward *u* (dotted) and northward *v* (solid) wind components delivered by conventional radiosonde measurements (twice daily) released on 26-28 December 2017 at the Wuhan weather station (~23.4 km away from our lidar site). Also shown are the corresponding profiles of the (d) range-corrected signal *X* and (e) volume depolarization ratio $\delta_v$ measured by the 355-nm polarization lidar on 26-27 December 2017. The different curve colors in each panel represent the radiosonde release times, as shown in Figure 3c. Each colored lidar profile represents a 1-h integration centered at the radiosonde release time marked in Figure3d. The radiosonde profiles quantitatively present the meteorological conditions that are pertinent to the warm front cloud at different stages and during precipitation.

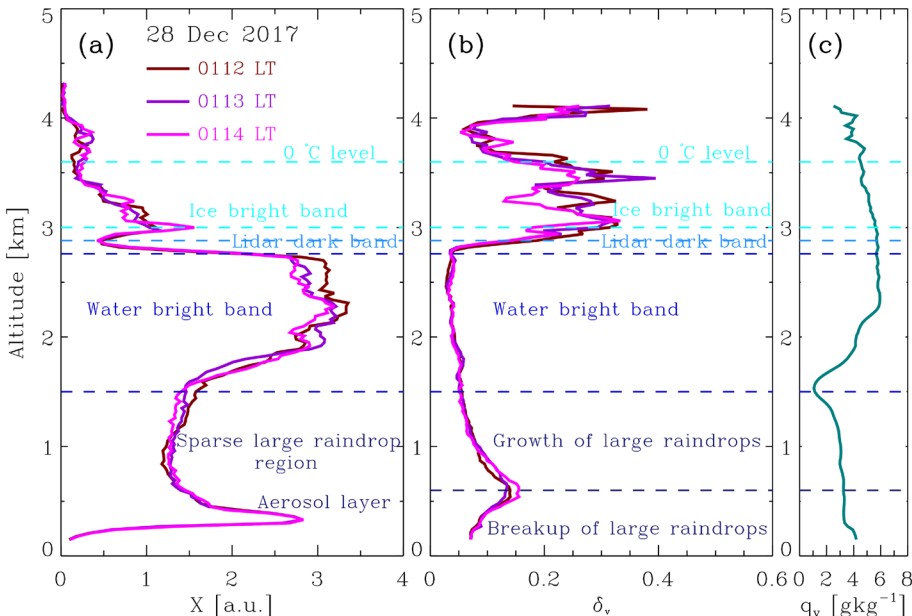

**Figure 4:** Lidar profiles for weak precipitation that reached the surface (drizzle). (a), Range-corrected 355-nm signal *X* profiles from 0112 to 0114 LT on 28 December 2017; (b), 355-nm volume depolarization ratio $\delta_v$ profiles in the same period; and (c), one-hour-averaged lidar $q_v$ profile centered at 0113 LT on the same day.

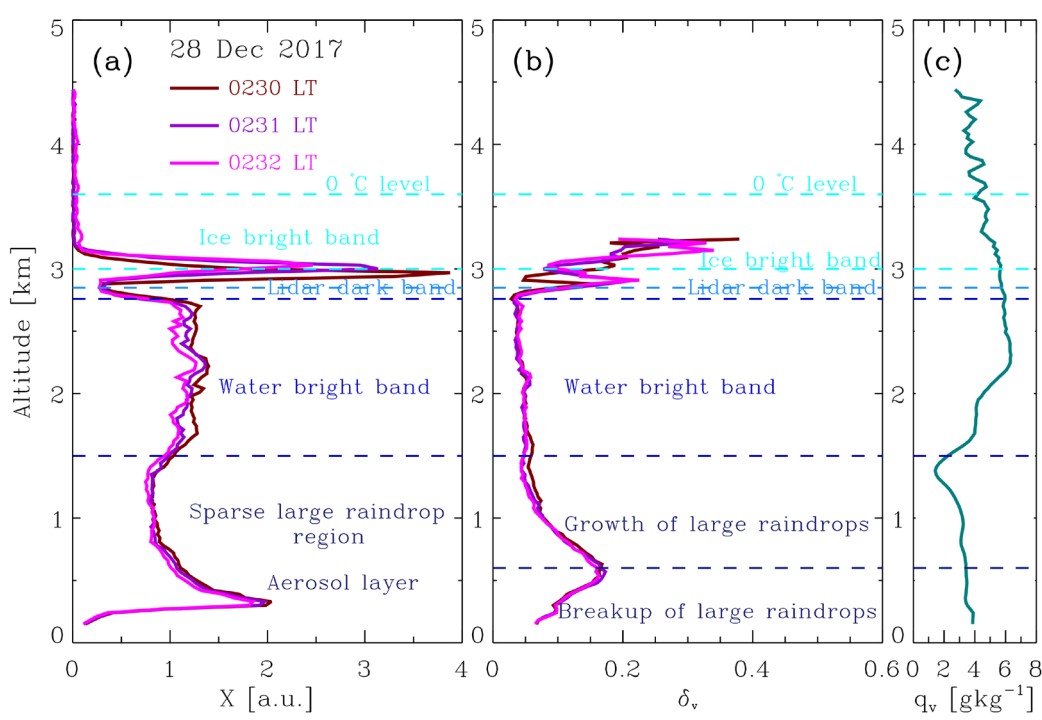

**Figure 5: Lidar profiles for slightly strong precipitation that reached the surface (in light rain). (a), Range-corrected 355-nm signal**
**X profiles covering the period from 0230 to 0232 LT on 28 December 2017;(b), 355-nmvolume depolarization ratio $\delta_v$ profiles**
**covering the same period;(c), one-hour-averaged lidar $q_v$ profile centered at 0231 LT on the same day.**


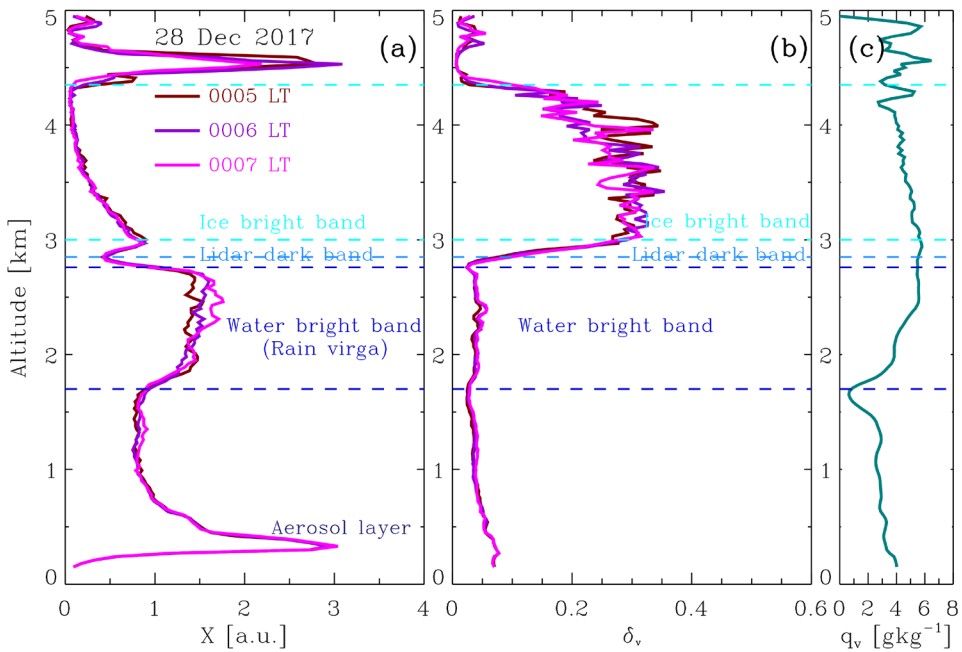

**Figure 6: Lidar profiles for a virga and its apparent source cloud occurring between intermittent reaching-surface precipitations.**
**(a), Range-corrected 355-nm signal *X* profiles from 0005 to 0007 LT on 28 December 2017; (b), 355-nmvolume depolarization ratio**
$\delta_v$ **profiles in the same period; and (c), one-hour-averaged lidar $q_v$ profile centered at 0006 LT on the same day.**



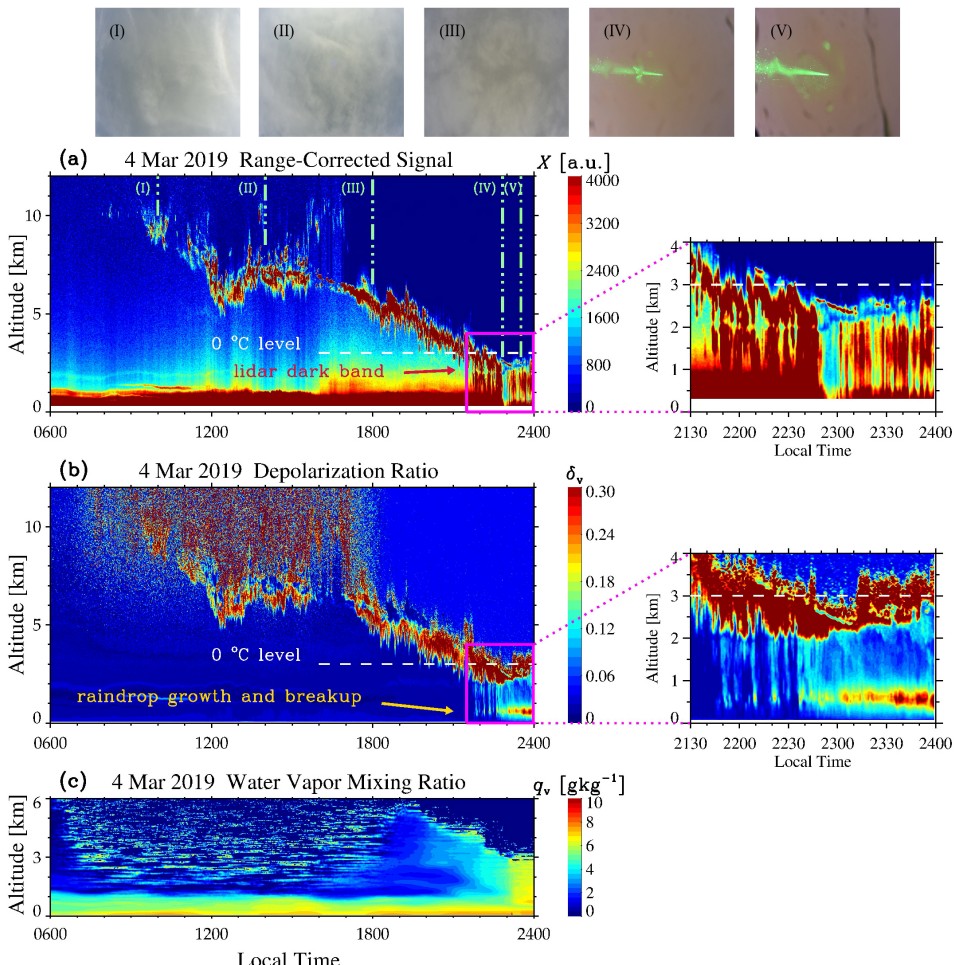

**Figure 7: Lidar observations (contour plots with 1 min/30 m resolution) of a moderate warm-front precipitation event. (a), Range-corrected 355-nm signal $X$; (b), 355-nm volume depolarization ratio $\delta_v$; and (c), water vapor mixing ratio $q_v$ (a sliding average of 60 min was applied). The precipitation streaks surrounded by magenta lines are zoomed in to show their details. Shown on the top of the figure are corresponding photographs of the sky taken by a ground-based camera at our lidar site, with the last two photographs exhibiting the rainy sky illuminated by a 532-nm laser beam.**


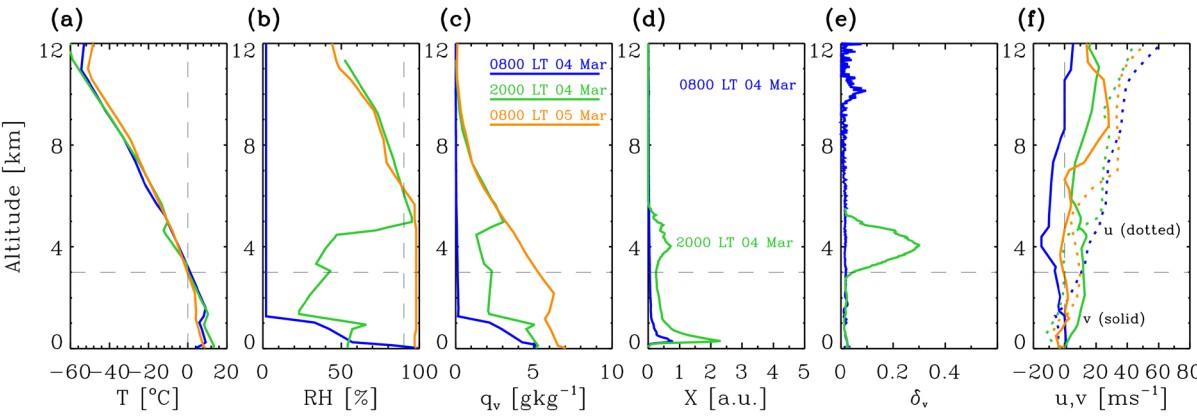

**Figure 8:** Sequential profiles of (a) temperature *T*, (b) relative humidity over liquid water *RH*, (c) water vapor mixing ratio *q_v* and (f)the eastward *u* (dashed) and northward *v* (solid) wind components delivered by conventional radiosondes (twice daily) released on 4-5 March 2019 at the Wuhan weather station. The corresponding profiles of the (d) range-corrected signal *X* and (e) volume depolarization ratio $\delta_v$ measured by the 355-nm polarization lidar on 4 March 2019 are also shown. The different curve colors in each panel represent the radiosonde release times, as shown in Figure 8c. Each colored lidar profile represents a 1-h integration centered at the radiosonde release time shown in Figure 8d. The radiosonde profiles quantitatively present the meteorological conditions pertinent to the warm-front clouds and precipitation.

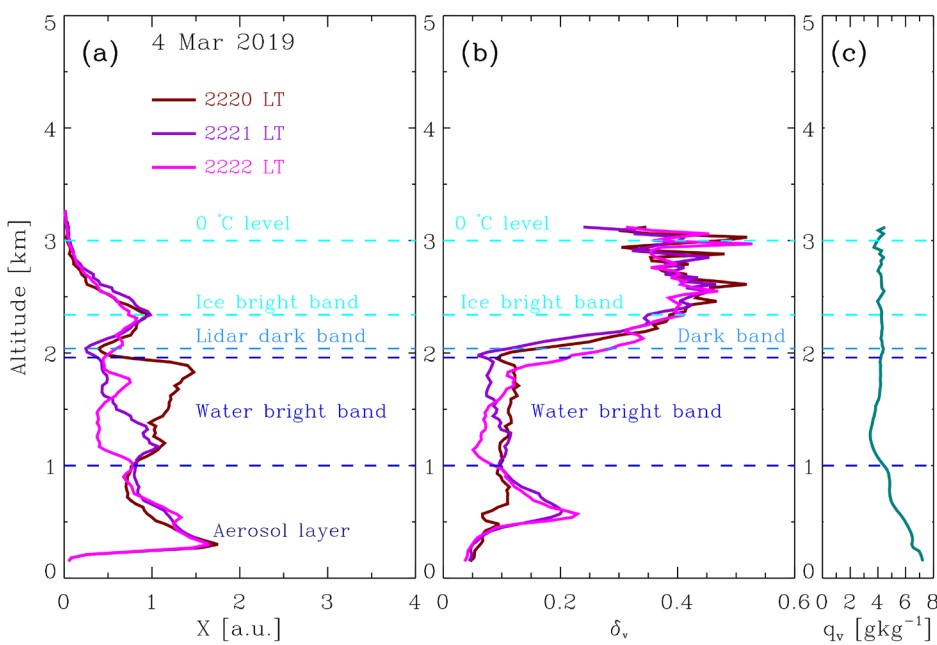

**Figure 9: Three 1-min lidar $X$ and $\delta_v$ profiles obtained from 2220 to 2222 LT on 4 March 2019 and a one-hour-averaged lidar $q_v$ profile centered at 2221 LT on the same day; the profiles, exhibit the vertical structure of the $X$ and $\delta_v$ precipitation streaks as well as the water vapor mixing ratio at the onset of the moderate warm-front precipitation event.**





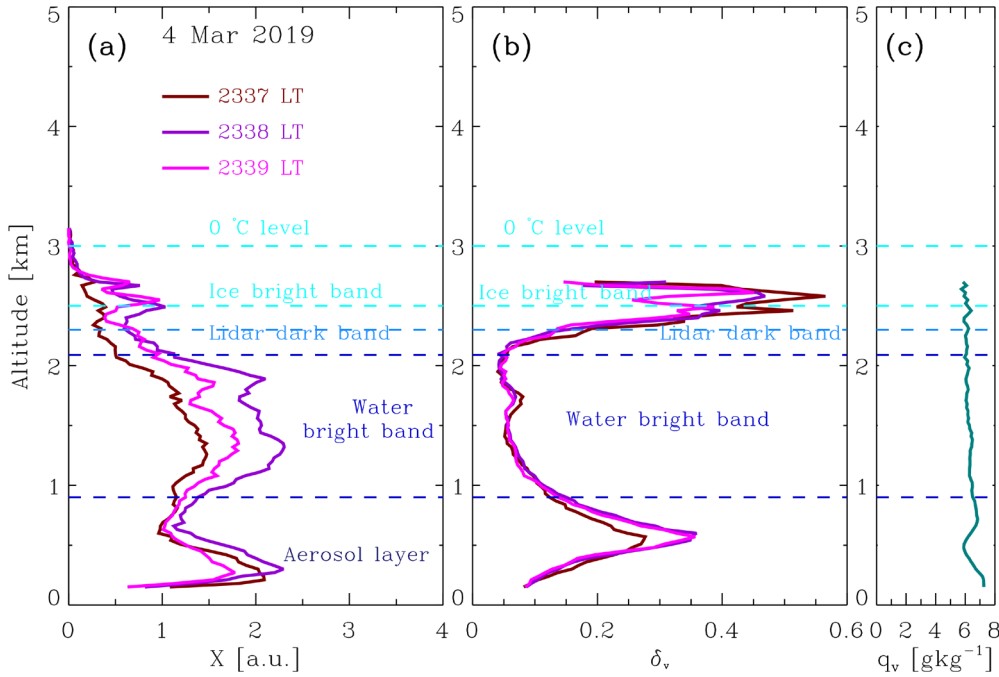

Figure 10: Three 1-min lidar $X$ and $\delta_v$ profiles covering the period from 2337 to 2339 LT on 4 March 2019 and a one-hour-averaged lidar $q_v$ profile centered at 2338 LT on the same day, exhibiting the vertical structure of the $X$ and $\delta_v$ precipitation streaks as well as the water vapor mixing ratio when the surface precipitation rate was highest (3.2 mm h[-1]) during the studied moderate warm-front precipitation event (yielding thick liquid water accumulation on the roof windows of the lidars; see photo V in Figure7).

## Appendix A: Discrimination criteria of spherical and nonspherical particles based on volume depolarization ratio

Here we derive the equivalent results expressed by the volume depolarization ratio $\delta_v$ based on the discrimination criteria of spherical and nonspherical particles given by the particle depolarization ratio $\delta_p$. The particle depolarization ratio $\delta_p$ can be obtained from the following equation (Cairo et al., 1999):

$$\delta_p(z) = \frac{(1 + \delta_m)\delta_v(z)R(z) - (1 + \delta_v(z))\delta_m}{(1 + \delta_m)R(z) - (1 + \delta_v(z))}, \quad (A1)$$

where $\delta_m$ is the molecular depolarization ratio and $R(z)$ the lidar backscatter ratio. In light of the theoretical calculation by Behrendt and Nakamura (2002), the $\delta_m$ value is ~0.004 for our 0.3-nm bandwidth polarization lidar (355 nm). Because the molecular depolarization ratio ($\delta_m(0.004) \ll 1$) can be neglected, the Eq. A1 is reduced to the following form

$$\delta_p(z) = \frac{1}{1 - \frac{1+\delta_v(z)}{R(z)}} \delta_v(z). \quad (A2)$$

According to Eq. A2, we can respectively derive the discrimination criteria of spherical and nonspherical particles expressed by the lidar-measured volume depolarization ratio $\delta_v$ based on those defined by the particle depolarization ratio $\delta_p$. In light of the previous observations, particles with $\delta_p < 0.1$ can be discriminated as spherical particles (Intrieri et al., 2002; Ansmann et al., 2008) and particles with $\delta_p > 0.2$ can be unquestionably discriminated as nonspherical particles (Wang and Sassen, 2001). In terms of Eq. A2, the discrimination threshold value of spherical particles takes the form

$$\frac{1}{1 - \frac{1+\delta_v(z)}{R(z)}} \delta_v(z) = 0.1, \quad \text{i.e.,} \, \delta_p(z) = 0.1, \quad (A3)$$

which is equivalent to

$$\delta_{v,\text{threshold}}(R) = 0.1 - \frac{0.11}{R + 0.1}. \quad (A4)$$

The lidar backscatter ratio $R$ has a theoretical value range of $[R_{\min}, \infty)$ with $R_{\min}$ being the minimum possible value of $R$ for the interested clouds/virgae (e.g., $R_{\min} = 7$ for the precipitation-related virgae). The corresponding $\delta_{v,\text{threshold}}$ has a value range of $\left[0.1 - \frac{0.11}{R_{\min}+0.1}, \, 0.1\right)$. Hence, the discrimination criterion of spherical particles expressed by $\delta_v(z)$ has the following form

$$\delta_v(z) < 0.1 - \frac{0.11}{R_{\min} + 0.1}. \quad (A5)$$

Inserting $R_{\min} = 7$ (for the precipitation-related virgae) into Eq. A5, we have $\delta_v(z) < 0.085$. This $\delta_v$ threshold value (0.085) of spherical particles is close to that value of 0.9 from the strict calculation based on Eq. A1. When $R_{\min} = 5$, we have $\delta_v(z) < 0.078$.

In terms of Eq. A2, the discrimination threshold value of nonspherical particles is given by

$$\frac{1}{1 - \frac{1+\delta_v(z)}{R(z)}} \delta_v(z) = 0.2, \quad \text{i.e.,} \, \delta_p(z) = 0.2, \quad (A6)$$

which is equivalent to

$$\delta_{v,\text{threshold}}(R) = 0.2 - \frac{0.24}{R + 0.2}. \quad (A7)$$

The lidar backscatter ratio $R$ has a theoretical value range of $[R_{\min}, \infty)$ with $R_{\min}$ being the minimum possible value of $R$ for the interested clouds/virgae (e.g., $R_{\min} = 7$ for the precipitation-related virgae). The corresponding $\delta_{v,\text{threshold}}$ has a value range of $[0.2 - \frac{0.24}{R_{\min}+0.2}, 0.2)$. Since $\delta_{v,\text{threshold}}(R)$ is a slowly-varying function of $R$ as seen from Eq. A7 (particularly when $R_{\min} \geq 5$), the discrimination criterion of nonspherical particles expressed by $\delta_v(z)$ can be written approximately as

$$\delta_v(z) > 0.2 - \frac{0.24}{R_{min} + 0.2}. \qquad (A8)$$

When $R_{min} = 7$, the discrimination criterion of nonspherical particles is given by $\delta_v(z) > 0.167$, which is equivalent to $\delta_p(z) > 0.2$ approximately.

In conclusion, the particle depolarization ratio $\delta_p$ has a quasilinear dependence on the volume depolarization ratio $\delta_v$, and a very weak dependence on lidar backscatter ratio $R$ (when $R \geq 5$). This favorable functional dependence allows us to utilize $\delta_v$ in discriminating whether the dominant lidar backscattering is attributed to spherical or nonspherical particles in a given backscatter volume. If $R_{min}$ is the minimum of the $R$ value range for interested clouds/virgae (e.g., $R_{min} = 7$ for the

precipitation-related virgae), the discrimination criterion of spherical particles expressed by $\delta_v(z)$ is given by Eq. A5, while the discrimination criterion of nonspherical particles expressed by $\delta_v(z)$ is given approximately by Eq. A8.