# Peer review of "Microphysical process of precipitating hydrometeors from warmfront mid-level stratiform clouds revealed by ground-based lidar observations"

_Atmospheric Chemistry and Physics, 2021_

## Referee Comment (RC3)

**General**

The paper contains interesting lidar observations of virga below mixed-phase clouds and a detailed, however often speculative interpretation of the virga observations. My overall impression is that the paper in its present form is not in a good shape. The manuscript is also quite long and should be shortened.

First of all, the lidar setup (zenith pointing, large receiver FOV of 1 mrad) can lead to very low depolarization ratio values introduced by specular reflection (because of zenith pointing). Furthermore, the relatively large FOV can sensitively influence the depolarization ratio observations via multiple scattering by droplets. Nothing is mentioned to these instrumental influences. The interesting finding of a local maximum of the depolarization ratio around 600 m height is in the near-range of the lidar so that systematic instrumental effects cannot be excluded. I mean, the overlap profile (incomplete overlap between laser beam and receiver FOV in the near range) is usually not well known and can vary with time (and diurnal cycle). All this causes artefacts. Nothing is mentioned to this problem.

The observations concentrate on the virga zone below the main altocumulus layers. More precise, the investigation is mainly focusing on the part of the virga from the 0°C level towards the surface. Nothing is said, why the ice crystals can survive such a long time at heights below the 0°C height level. Obviously, the melting crystals cool the surrounding air and keep the temperatures close to 0°C, even up to 1 km below the 0°C height level where the radiosonde (obviously ascending outside of virga) indicated temperatures up to 6°C.

Nothing is mentioned, why raindrops lead to enhanced depolarization ratios! Maybe I overlooked it. Is that because the shapes of the rain drops are no longer spherical during falling, the shape is like the one of pears with the flat side in falling direction…? Please explain!

Nothing is mentioned about the impact of multiple scattering in layers with high droplet concentration. Maybe the raindrop maximum at 600 m is just caused by multiple scattering by numerous small droplets and not by 'a few big', nonspherical rain drops. In the paper, there are no profiles of particle backscatter and extinction coefficients. So, there is no opportunity to conclude on the multiple scattering effect. I was puzzled by the fact that the depolarization maximum was always around 600 m height, why not at 800 or 400 or at 1000~m height? I speculate that may have to do with systematic instrumental problems in the near-range of the lidar.

The most important point of concern is the following: I have a severe problem with the 'theory' of the authors how ice crystals are nucleated via heterogeneous ice nucleation. The authors believe that large (1mm in size) droplets fall out of an altocumulus layer and then they immediately freeze right below cloud base. I have never heart about such an ice nucleation process. Furthermore, I asked myself: How can 1 mm droplets form in an altocumulus cloud layer where typical droplet sizes are 10-20µm …)

The established, common 'theory' of ice nucleation in altocumulus layers is the following:

Ice nucleation (dominated by immersion freezing for temperatures >-25°C) starts at the coldest point of the cloud, i.e., at cloud top. At cloud top the probability of ice nucleation is largest because ice nucleation is a strong function of temperature. The probability increases by an order of magnitude when temperature decreases by 5K. Thus in cases of 500-600m

thick cloud layers the ice nucleation probability at cloud top is an order of magnitude higher than at cloud base.

So, most probably first ice crystals nucleate at cloud top via immersion freezing (liquid water droplets freeze), in the liquid-water droplet environment of the altocumulus layer. Then, in the next step, these ice crystals grow fast and immediately start falling. They grow to about 100µm within 60-120s! During falling they continuously grow as long as ice supersaturation is given. When ice subsaturation levels are reached the ice crystals start to shrink and to evaporate. When entering the air mass below the 0°C height level the crystals start to melt but during this process they consume so much energy that they are able to keep the temperatures of the ambient air close to 0°C (in your cases down to almost 1000m below the 0°C height level) although the radiosonde may have measured 6°C.

Finally, the manuscript tells us nothing about the true ice nucleation. There is no information about the altocumulus layer (e.g. cloud top height and temperature), there is no information and discussion about potential seeder/feeder effects (no information about ice cloud layers above the altocumulus), and there is no information about secondary ice formation (triggered by the Hallett Mossop effect) dominating in the height range in which the temperatures are between -5 and -8°C. All this influences the virga properties that are exhaustingly discussed in the paper. It should be clearly said in the introduction that the ice nucleating processes are not covered by the paper. The paper exclusively focus on the virga.

Many aspects mentioned above (but not discussed in the manuscript) triggered many questions! Please do not misunderstand me! I like the approach and want to help to improve the paper. I know that readers appreciate if the authors are critical to their own observations!

Major revisions are needed.

**Details:**

P1, l18: surface rainfall…..? is not just self-explaining: better use: precipitation that reached the surface. You may define 'surface rain' in the introduction, but I personally do not like such wording.

P1, l19: parent cloud …? Is also not self-explaining: better: …falling out of a shallow mixed-phase cloud layer. Why do we need such a wording? And in the case of seeding from above, then we have grandparent clouds…? I would just call or denote such a clouds … shallow cloud layer or altocumulus layer.

Please change this (surface rain, parent cloud) throughout the article.

P1, l55-56: I think your hypothesis is wrong: …suggesting that most supercooled liquid drops falling out of parent clouds rapidly froze into ice crystals on the tops of virga. See my explanation above.

P3,l83: The lidar is pointing exactly vertically! That means you may have very low depolarization ratios from specular reflection by falling, oriented ice crystals. And layers with specular reflection may be misclassified as liquid-droplet layers.

P3, l84: The receiver field of view is 1 mrad. This means you may have considerable problems with multiple scattering. Multiple scattering in environments with droplets can

cause significantly enhanced depolarization ratios. And these high depolarization ratios may be interpreted as ice crystals or as big rain drops. So, there is room for ambiguous interpretation.

That should be commented (zenith pointing, specular reflection, multiple scattering, enhanced depolarization, overlap impact, signal gluing impact).

P4, l93: Please state in which height range gluing of signal profiles is performed (for the cases discussed in Sect. 3.)

P4, l104: What about multiple scattering in water-droplet layers, and corresponding increase in depolarization ratios? Please comment on that!

P4, l111: Below the dark band, enhanced depolarization ratio indicates rain drops. Please explain why? The shape of rain drops during falling deviates from the perfect spherical form? They look like pears? With the flat surface into the falling direction? Please comment on that.

P5, l144: An explanation is need why ice crystals can survive as ice crystals over a distance of 600 m below the height of 0°C level, i.e., for about 1200 s (20 min) when falling speed is high with 50 cm/s. To my opinion, cooling of the surrounding air during evaporation and melting… is the reason.

P6, l 171, 172, 173: parent cloud… I do not like this wording. Furthermore, you do not know whether the crystals were formed in that cloud layer. Maybe ice crystals from above seeded the cloud. So, please be careful with the argumentation. Limit the argumentation to topics and facts that were observed.

P6, Sect 3.3.1: I found this section is too long.

P7, Sect 3.1.2: Check to what extent multiple scattering effects and specular reflection could have influenced the observations.

P8, l218: Why is the volume depol ratio not close to 0.01? Maybe be because of multiple scattering. The paper does not contain any backscatter and extinction coefficients. So, I have no idea whether multiple scattering could be problem or not.

P8, l229-230:  How can droplets evaporate and at the same time others grow….?

P8, l235: ..Peaks at 0.1-0.4?  …you mean 0.1-0.14? Why is the rain depol peak always at 600 m. Can we have an estimate for the backscatter and the extinction coefficient? Maybe multiple scattering had an influence!

P9, l240-l244: To my opinion, this is speculation. …should be avoided.

P9, l269: Large values of the range - corrected signal coinciding with low depol ratio! That could have been caused by specular reflection by a few falling and oriented ice crystals. One should discuss such influences.

P9, l279-295: What about seeding by clouds higher up. You have no information about all this. There is so much speculation here. Please avoid that. Keep the discussion short.

P10, l297-308: Here you present again the erroneous ice nucleation theory! You should at least also present the established one (in the absence of seeding from above, ice crystals nucleate at cloud top, grow fast and start falling through the cloudoand become large before they leave the main cloud layer and show up in the virga as quite large crystals that may further grow as long a supersaturation over ice is given.

The entire Sect. 3.1.2 is very long and contains many speculative statements. The entire section should be shortened and should be based on what was observed.

P11, Sect 3.2: Another case, again depol peak caused by rain at 600 m! Why always at 600 m height? Should be clarified and discussed.

Speculation about break up processes, occurrence of ensembles of small and large droplets, collision-coalescence effects… all this sounds convincing, but is that the truth? … As long as the role of multiple scattering is not clarified, the discussion is not trustworthy.

Sect. 3.2.1: There is no information about the clouds, the ice nucleation processes, potential secondary ice nucleation processes, seeding effects, nothing. That should be emphasized to keep the entire discussion short. You should concentrate on the virga, because more is not possible. That must be clearly stated in the manuscript.

P12, l369: Ice crystals detected 960m below 0°C height level…. Again, how can they survive? You have to give a reason!

Sect. 3.3.2: ..again, what is the potential impact of multiple scattering?

Figures:

Figure 1: The colored figures should be shown from 27 Dec, 00:00 LT to 28 Dec 16 LT, and probably up to 7 km only to see the necessary details. In the present form, the figure is almost useless.

Figure 3: Is cloud top at 4 km height? We do not know! The decreased depolarization ratio from 3.6 to 4 km could be caused by specular reflection? Who knows! The decrease of the depolarization ratio below the local maximum at 600 m height  may be an instrumental effect (bias) because the observations are performed in the near-range of the lidar where nothing is well defined. Please comment on that! So break up of rain droplets is speculation. Also evaporation could have started and the droplets got smaller and spherical again…

Figure 4:  Again the overlap problems in the lowest 500 m! Can we trust the depolarization ratio values at heights  below 500 m?

Figure 5: Again a layer with low depolarization ratio around 4.5 km height together with large signal! Is that the liquid cloud layer? Or are there falling, oriented crystals producing specular reflection? This time no enhanced depolarization ratio around 600 m height, no big raindrops? This case demonstrates that there is slight decrease of the depolarization ratio from 250 m to 1 km. This is probably the background depol height profile in the absence of any multiple scattering effect.

Figure 6: Bad quality, like Figure 1: What do you want to show. There is almost nothing to see!

Figure 8: Again, one sees the high depolarization ratios indicating ice crystals, one does not see the main cloud layer. One does not know whether the cloud was seeded by higher clouds. One does not see anything. Except the virga of ice crystals, and then the drop in the depolarization ratio, when the crystals melt. Around 600 m again, rain droplets! Always around 600 m! What is the possible reason that the depolarization ratio maximum is always at about 600m. Maybe caused by  multiple scattering?

Figure 9 again: Depolarization maximum at 600 m! Why always at 600 m?

Again, please use all of my comments as a constructive contribution to improve the paper. The topic of the article is interesting and deserves publication!

---

## Author Comment (AC1)

**Response to reviewer 1 (RC1, Anonymous Referee #1)**

Reviewer's comments are presented here by italics

***Comments:***

*1) general comments*

*The paper deals with lidar measurements to improve the understanding of microphysical process of mid-level stratiform clouds. The results of this study are based on two case studies observed in 2017 and 2019. The authors also highlight, that lidar observations of precipitating cloud systems where the whole precipitation process can be studied are rare but needed to understand the process from origin till rain hits the ground*

*The data are obtained by two lidar systems, a depolarization and a water vapor Raman lidar. The radar systems are designed to be able to measure also during light rain – optics of the systems are protected by a glass window in the roof of the institute.*

*The measurements depict two warm front cloud systems overpassing the measurement site. These lidar observations are described and related to precipitation formation processes. While the liquid microphysical processes seam to dominated the analysis.*

*Generally, the structure in the paper is not clear enough. The result section 3 is missing a red line to follow. It might be helpful to make more paragraphs and structure them better. It is not always easy to connect the information with the actual microphysical processes observed. So having more explanation of what process is happening and explain the resulting observation signatures would help. Perhaps use a sematic sketch? If this could be improved the quality of the paper would rise for sure.*

**Authors' response:**

The authors greatly appreciate this reviewer for his affirmative remark, constructive criticism and kind suggestions. Taking all the comments into account, the manuscript has been revised. In particular, more paragraphs to explain the observed results have been added, and they are carefully structured in the revised manuscript. With respect to the schematic sketch to explain the lidar-observed results, please allow us to do this in future (the lidar and disdrometer observations are continuing at our site) because the authors' capability to make a satisfied artwork appears to be immature at present.

***Comments:***

*2) specific comments*

*•Section 2.1. line 73-75 and section 2.1.1 line 111-114*

*Are there data or plots available to show the results of the water splashing experiment? From my site the performed technique is new, so results of it should be presented or at least citations given to similar performed experiments.*

**Authors' response:**

Taking the reviewer's suggestion, an example about the results of the water splashing experiment has been given in the revised manuscript (please see Figure 1 in the revised manuscript). The pertinent description (section 2.1.1 lines 111-114 in the previous manuscript) has changed to "An artificial water splashing experiment was performed on the lidar roof windows to examine the effects of water accumulation. A comparison of the lidar profiles with and without water accumulation on the lidar roof windows is given in Figure 1. Enhanced lidar signal ($X$) and depolarization ($\delta_v$) values at altitudes around 4.0 km resulted from an optically-thick (opaque) water-droplet cloud layer because there existed a high $X$ value and near-zero $\delta_v$ value (~0.008) on the cloud base (~3.9 km) (Wang and Sassen, 2001), and also there initially existed a monotonic rapid increase in both the values of $X$ and $\delta_v$ with increasing penetration of laser light into the layer. The cloud-related structures shown in both the $X$ and $\delta_v$ profiles were consistent before and after water splashing (particularly, cloud base altitudes). This comparison clearly shows that water accumulation on the lidar roof windows yielded nearly height-independent lidar signal ($X$) attenuation, and neither the cloud-related $X$ vertical structure nor the profile of the volume depolarization ratio $\delta_v$ (the magnitude and vertical structure) were altered. This result is physically reasonable." (please see lines 128-138)

[Figure]

**Figure 1: Comparison of the lidar profiles with (integrated from 2030 to 2032 LT on 31 May 2020, dashed blue line) and without (integrated from 2028 to 2030 LT on the same day, solid red line) water accumulation on the lidar roof windows. (a), Range-corrected 355-nm signal $X$ profiles; (b), 355-nm volume depolarization ratio $\delta_v$ profiles. The water accumulation was produced by the artificial water splashing experiment.**

Citation added

Wang, Z., and K. Sassen: Cloud type and macrophysical property retrieval using multiple remote sensors, J. Appl. Meteorol., 40(10), 1665–1682, https://doi:10.1175/1520-0450(2001)040<1665:CTAMPR>2.0.CO;2, 2001.

*Comments:*
*2) specific comments*
*•Section 2.1.1 line 103-108*
*The explanation of the dark band is hard to follow. Could you split the sentence into two or 3 parts and extend the explanation a bit so that it is better to read?*

**Authors' response:**

In light of the reviewer's suggestion, the relevant sentences (section 2.1.1 lines 103-108 in the previous manuscript) have been revised as "The magnitude of the $\delta_v$ value allows us to identify whether the dominant backscattering is attributed to ice crystals or water droplets in a given backscatter volume (Shupe, 2007). In general, liquid water droplets suspended in the atmosphere are nearly spherical and produce a very low depolarization ratio (close to zero) for single scattering at exact 180°, while ice crystals, which are usually nonspherical, generate a quite large depolarization ratio in the 180° backscattering direction. For some mid-level stratiform precipitations, gravitationally-falling hydrometeors form initially at altitudes above the 0 °C isotherm level. They fall often as mixed-phase hydrometeors (supercooled liquid drops and ice crystals/snowflakes) at sub-zero temperature during their early descent. After the falling mixed-phase hydrometeors pass through the 0 °C isotherm level, the snowflake (ice)-to-raindrop transition can yield a shallow layer of relatively smaller lidar echoes (a local $X$ minimum), that is called "lidar dark band" (Sassen and Chen, 1995; Di Girolamo et al., 2012). The lidar dark band can be used to differentiate between the altitudinal regions with ice-containing particles above the dark band and pure liquid raindrops below the dark band. Hence, at altitudes above the dark band, the discrimination criteria in terms of the depolarization ratio magnitude are $\delta_v < 0.1$ for water droplets/drops and $\delta_v > 0.2$ for ice crystals (Intrieri et al., 2002; Shupe et al., 2008), while an enhanced depolarization ratio ($\delta_v > 0.1$) at altitudes below the dark band indicates the presence of large raindrops." (please see lines 103-116)

*Comments:*
*2) specific comments*
*•Section 3.1. Figure 1*
*The text below the Figure is too long. Describe what the graphs show, do not give any interpretation or highlight things the graphs show the caption. All interpretations or highlights that can be seen have to be in the main text of the article.*

**Authors' response:**

Taking the reviewer's suggestion, the caption of Figure 2 (Figure 1 in the previous manuscript) has been shortened as "Figure 2: Time-height contour plots (1 min/30 m resolution) of the (a) range-corrected signal $X$, (b) volume depolarization ratio $\delta_v$ measured by a 355-nm polarization lidar, and (c) water vapor mixing ratio $q_v$ measured by a water vapor Raman lidar on 26–28 December 2017, which exhibited the passage of a warm front and the resulting hours-long light rain. A sliding average of 60 min was applied to the Raman lidar data. The precipitation streaks surrounded by magenta lines are zoomed in to show their details. Shown on the top of the figure are

the corresponding photographs of the sky taken by a ground-based camera at our lidar site, with the third photograph exhibiting the sky illuminated by a 532-nm laser beam during the onset of rainfall." (please see lines 660-665). The interpretations and highlights have been moved to the main text.

***Comments:***
*2) specific comments*
*•Section 3.1.1 sentence line 160-160 and following sentences*
*I had a hard time to follow the text here and connect the information you give to the story you want to tell. Please structure this paragraph clear. What can be seen in the graph and what do you follow from your observations. Perhaps make some paragraphs to give the text more structure.*

**Authors' response:**

The authors sincerely appreciate the reviewer for his kind suggestion. In light of this suggestion and a comment ("I found this section is too long") from another reviewer (reviewer 3), the text in subsection 3.1.1 has been reorganized as follows.

"Figure 3 presents the radiosonde profiles that are pertinent to the warm-front cloud at different stages and during precipitation, together with the 1-h mean lidar profiles obtained during the radiosonde launches. The temporally-varying cloud properties (e.g., falling cloud base, increasing cloud thickness and variable cloud types) between 2000 LT on 26 December and 2000 LT on 27 December 2017 coincided with the classical picture of preceding upglide clouds of an advancing warm-front system. Accordingly, a downgoing moist layer was observed strengthening and broadening with time during this period (Figs. 3b and 3c). At the cloud base (except cirrus), the relative humidity over ice had values close to the relative humidity threshold of 84% that is conventionally used to determine the cloud base heights (Wang and Rossow, 1995; Zhang et al., 2018). Furthermore, the radiosonde data exhibited that the southwesterly wind mostly prevailed at the cloud altitudes (Figs. 3d, 3e and 3f), and the air pressure at altitudes of ~0–5 km dropped continuously by ~3–5 hPa in the period (not shown here), which did belong to the typical warm-front features.

The radiosonde released at 0800 LT on 28 December 2017 provided measurements of the meteorological conditions when precipitation reached the surface, although the lidar measurements had already terminated (at 0538 LT) ~ 2 hours earlier. As seen from Figure 3b (red), the relative humidity reached a maximum of 98% with respect to water in an altitude range of ~3–4 km, immediately above the tops of the liquid precipitation streaks (at ~3 km, see Figs. 2a and 2b). Water vapor at altitudes of ~3–9 km was advected from the southwest, as seen in the wind component profiles (Figure 3f, red). The high water vapor mixing ratios observed at altitudes below ~3 km came from the evaporation of falling raindrops." (please see lines 182-198)

***Comments:***
*2) specific comments*
*•Section 3.1.2 line 201-204*

*This explanation has to be given when you explain the water splashing experiment! So move this up in the section above!*

**Authors' response:**

The explanation has changed to "Although the rainfall-induced water accumulation on the roof window of the lidar varied with time, the precipitation streaks and dark band were steadily reasonably displayed in the $X$ and $\delta_v$ time-height plots (Figs. 2a and 2b). This is consistent with the result of our water splashing experiment." in the revised manuscript (please see lines 211-214).

***Comments:***

*2) specific comments*
*Section 3.1.2 line 210-211*
*Can you explain these in more detail or give a citation? Is there a relation to the signature and the distance to the 1km or higher origin layer of the initiation? Can signatures be used to identify the high of initiation?*

**Authors' response:**

Taking the reviewer's suggestion, we have added the following explanations and citation in the revised manuscript.

 "Note that the formation of gravitationally-falling ice-containing hydrometeors requires ambient temperatures colder than $-4°C$ (Rangno and Hobbs, 2001; Yi et al., 2021) and ice nucleation (via contact freezing) is active at temperatures around $-10°C$ (Ansmann et al., 2008). Given a mean lapse rate of $6.5°C$ $km^{-1}$, it is expected that the mixed-phase stratiform precipitations would begin at altitudes more than 1 km above the $0°C$ isotherm level. In fact, the existing cloud/precipitation radar observations also have indicated that the stratiform precipitations with the snowflake-to-raindrop transition (where the radar bright band occurs corresponding to the lidar dark band) initiate usually at altitudes more than 1 km above $0°C$ isotherm level (e.g., Di Girolamo et al., 2012, Fig. 2; Pfitzenmaier et al., 2018, Fig. 4)." (please see Subsection 3.1.2 lines 221-228)

Citation added

Rangno, A. L. and Hobbs, P. V.: Ice particles in stratiform clouds in the Arctic and possible mechanisms for the production of high ice concentrations, J. Geophys. Res., 106(D14), 15,065-15,075, https://doi:10.1029/2000JD900286, 2001.

Yi, Y., Yi, F., Liu, F., Zhang, Y., Yu, C., and He, Y.: A prolonged and widespread thin mid-level liquid cloud layer as observed by ground-based lidars, radiosonde and space-borne instruments, Atmos. Res., 263, 105815, https://doi.org/10.1016/j.atmosres.2021.105815, 2021.

***Comments:***

*2) specific comments*
*•Section 3.1.2 paragraph 4 (234-243)*
*Can you explain this in more detail? Are there other observations done showing the same, give a citation? It would be nice to get a bit more explanation for people not so familiar with lidar measurements.*

**Authors' response:**

In light of the reviewer's suggestion, with respect to this finding, more explanations have been given and some statements about its novelty have been made in revised manuscript (In brief, our lidar observations reveals for the first time (to our knowledge) the collision-coalescence growth and subsequent spontaneous breakup of falling raindrops, that actually take place in the natural atmosphere. They represent the posterior microphysical processes necessary for the surface rain production.). Thus, the subsection 3.1.2 paragraph 4 has changed to

"At altitudes below the water bright band, the precipitation-related lidar backscattering ($X$) apparently weakened (Fig. 4a, in which the enhanced $X$ values at altitudes from 0.3–0.7 km resulted from the boundary layer aerosols), indicating low-density raindrops there, whereas $\delta_v$ first increased with decreasing height and then decreased after reaching a maximum (0.13–0.16) at an altitude of approximately 0.6 km (Fig. 4b). Here the magnitude and altitude variation of the lidar depolarization ratio $\delta_v$ values allow us to identify where large-sized raindrops form and break up. Falling small-sized raindrops (equivalent diameter $\leq$ 1.0 mm) are quasi-spherical (Pruppacher and Klett, 1997) and yield small $\delta_v$ values (generally less than 0.1), whereas falling large-sized raindrops (equivalent diameter > 2.8 mm) become nonspherical (with flat or hollow bottom in falling direction) (Pruppacher and Klett, 1997) and lead to large $\delta_v$ values (larger than 0.1). In fact, prominent $\delta_v$ peaks (~0.1–0.4) at altitudes of approximately 0.6 km are always observed in the $\delta_v$ profiles related to reaching-surface precipitation in the present light rain case (Fig. 2). The $\delta_v$ maxima at an altitude of ~0.6 km are much larger than the typical values (less than ~0.07) observed by our 355-nm polarization lidar at approximately the same altitude during rainless days. Here we can exclude a possibility that the $\delta_v$ maxima (~0.1–0.4) at ~0.6-km altitude resulted from multiple scattering by dense droplets around this altitude. As mentioned above, for the 1-mrad receiver FOV, a dense water-droplet cloud layer with the multiple-scattering-induced depolarization ratio $\delta_v$ values larger than 0.1 is optically opaque. In contrast to this situation, in our case, when the prominent $\delta_v$ peak (~0.1–0.4) around 0.6-km altitude occurred, the vertical structure of the precipitation streaks at altitudes far above 0.6 km (e.g., ice bright band, lidar dark band and lidar water bright band) was unambiguously detected by our polarization lidar, indicating that the enhanced depolarization ratios around 0.6-km altitude cannot be caused by multiple scattering from dense spherical water droplets therein. Furthermore, since most falling raindrops evaporated and vanished in the liquid-water bright band as indicated by the enhanced water vapor mixing ratio therein and rapidly-decreasing lidar signal on the bottom of the water bright band, small droplets at altitudes below the water bright band were hardly dense enough to generate a strong multiple scattering with $\delta_v \geq 0.1$. Therefore, our observational results suggest that sparse large raindrops that fall out of the water bright band with higher fall velocities further grow in size by collecting smaller raindrops along their fall paths. They grow to sizes at which spontaneous breakup occurs at an altitude of approximately 0.6 km. In brief, our lidar observations reveal for the first time (to our knowledge) the collision-coalescence growth and subsequent spontaneous breakup of falling raindrops, that actually take place in the natural atmosphere. They represent

the posterior microphysical processes necessary for the reaching-surface precipitation production. Interestingly, the size maximization of falling raindrops as shown by the strongest nonspherical shapes (maximum depolarization ratio values) always appeared at an altitude of ~0.6 km for a variety of mid-level stratiform precipitations (in light of our observations). Obviously, the explanation to this ubiquitous feature needs further observational and modelling efforts. As seen in Fig. 2b, the boundary layer aerosols had little impact on the $\delta_v$ precipitation streaks. In addition, at altitudes below 1.5 km, the $q_v$ values decreased with increasing altitude, reflecting a normal altitude distribution of the boundary layer water vapor." (please see lines 249-279).

***Comments:***
*2) specific comments*
*•Section 3.1.2 line 306*
*Does this comparison make sense here? 1 mm large super cooled droplets? Could you comment on this please and give a reference!*

**Authors' response:**
Taking the reviewer's comments into account, the sentence about this comparison (Their freezing time would be 25–50 s given a falling velocity of 4 ms$^{-1}$ for ~1.0-mm liquid drops.) has been dropped in the revised manuscript.

***Comments:***
*2) specific comments*
*•Section 3.1.2. line 313-325*
*This pat was hard to follow. It might be one of the mature parts of the paper. Please, describe what you observed and in a second step what process might be behind. Perhaps it makes also sense to make a summarizing sketch of the processes observed and relate them to the measurements you would expect. Then it is easier to follow for the readers.*

**Authors' response:**
    The authors sincerely thank this reviewer for his kind suggestion. Following the suggested expression logic (first describing the observational results and then stating what process might be behind), the related sentences have been revised as

"The local depolarization minimum ($\delta_v \leq 0.04$) was persistently observed immediately beneath (~100 m below) the lidar dark-band minimum ($X$ minimum). This displayed that the completion of the melting process of most falling ice particles took place at altitudes (hundreds of meters) below the 0°C isotherm level. The liquid-water bright band (with a geometrical thickness of ~ 1 km) just below the lidar dark band was characterized by enhanced $X$ values and small $\delta_v$ values. There existed a high-concentration moisture (large $q_v$ values) in this bright band. These features indicate that the liquid-water bright band resulted from gravitationally-falling, dense evaporating liquid drops. In terms of the lidar-measured profiles during reaching-surface precipitation, at altitudes below the water bright band, the precipitation-related lidar backscattering apparently weakened, while $\delta_v$ first increased with decreasing altitude and then decreased after reaching a prominent maximum at

an altitude of ~0.6 km. The lidar profiles for the virgae showed narrower and weaker water bright bands than those observed when precipitation reached the surface. Moreover, during virga occurrence, there was no perceptible depolarization enhancement at an altitude of ~0.6 km. By combining the above-mentioned lidar observations, a picture on the microphysical processes of falling hydrometeors in liquid-phase stage emerged. After going through the dark band, most falling raindrops shrank or vanished in the water bright band due to evaporation, whereas a few large raindrops survived and fell out of the water bright band when the rain rate below the apparent source cloud base was high enough. The large raindrops might come from both the complete melting of large falling ice/snow particles and collision-coalescence formation in the dense water bright band. Sparse, large raindrops with high fall velocities further grew in size by collecting smaller raindrops along their fall paths. At an altitude of ~0.6 km, the large raindrops grew to the sizes at which spontaneous breakup could occur, yielding reaching-surface precipitation. When the rain rate below the apparent source cloud base was low, nearly none of the large raindrops fell out of the water bright band. Consequently, there were only virgae suspended on air (without reaching-surface precipitation)." (please see lines 352-371)

***Comments:***
*3) Technical corrections*
*•Line 216-218: Please reformulate this sentence*
**Authors' response:**
The sentence has been rewritten as

"A liquid water bright band appeared as a layer of relatively large particle backscatter values, located at ~1.50–2.76-km altitudes, just below the lidar dark band (Fig. 4). It is called "weak lidar bright band" in the literatures (Sassen and Chen, 1995; Di Girolamo et al., 2012)." (please see lines 233-235).

***Comments:***
*3) Technical corrections*
*•Please have a clearer structure in your sections and paragraphs*
**Authors' response:**
The structures in the sections and paragraphs have been reorganized in light of the specific suggestions from this reviewer.

***Comments:***
*3) Technical corrections*
*•Make more paragraphs*
**Authors' response:**
More paragraphs have been made in the revised manuscript in order to structure the sections clear.

***Comments:***
*3) Technical corrections*

*•Shorten you captions of the figures; some are quite long. Put the information into the text or make more figures*

**Authors' response:**

The captions of the figures have been shortened and pertinent information has been put into the text.

---

## Author Comment (AC2)

**Response to reviewer 2 (RC2, Anonymous Referee #3)**

Reviewer's comments are presented here by italics

***Comments:***
*The manuscript reported on the microphysical processes of precipitating hydrometers that occur at altitudes ranging from the parent cloud base down to the near surface for two warm-frontal precipitation episodes. The results are based on the simultaneously observed sequential profiles of the range-corrected signal X, volume depolarization ratio v and water vapor mixing ratio qv from the combination of a 355-nm polarization lidar and water vapor Raman lidar at Wuhan University atmospheric observatory. The observational period ranges clear-sky, cloud to precipitation, allowing for the process analysis. The observational result could potentially contribute much to the application of convection forecast. More importantly, something interesting is revealed for the first time for both light and moderation rainfall from warm-front mid-level stratiform clouds in Wuhan, Central China. To minimize of the water accumulation impact of lidar roof window on lidar signals, the authors conducted an artificial water splashing experiment, showing promising results for its height-independent lidar signal. The observational methodologies are novel, and the analysis seems scientifically sound to me. Therefore, findings are convincing and deserve rapid acceptance for publication at ACP. Nevertheless, fruitful discussion regarding the warm front is still lacking. Besides, there are other minor issues that need to be addressed prior to formal acceptance, which are listed as below:*

**Authors' response:**
  The authors sincerely thank this reviewer for evaluating this manuscript and clearly pointing out its novelty and significance as well as drawbacks. In light of the reviewer's suggestions, more discussions about the warm front have been added and the listed issues have been addressed in the revised manuscript.

***Major comments:***
*1. WARM FRONT: More discussion is required. Except for the prevailing wind, a variety of other meteorological variables can be used to characterize the warm front, including air pressure level, the temporally varying cloud properties. For instance, when a warm front is approaching, the barometric pressure begins decreasing, the wispy and high cirrus clouds appear. Then, the layer of clouds tends to thicken along with raindrop falls from the cloud base as it arrives, meanwhile, nimbus, cumulus, and stratus clouds can be observed. Generally, the precipitation associated with the warm front is light and steady, and thus its intensity is moderate, and can last for several days. Under some special conditions, the warm fronts also accounts for the thunderstorms and intense precipitation. Alternatively, to better illustrate of the two warm-front rainfall cases, the authors may take a look at the composite synoptic map showing surfacee-level geopotential height and surface potential temperature, in*

*which the warm front is supposed to be marked.*

**Authors' response:**

Taking the reviewer's suggestion and another reviewer's opinion (reviewer 3: Sect 3.1.1 is too long), some brief descriptions about the observed warm-front characteristics have been added in the revised manuscript.

"The temporally-varying cloud properties (e.g., falling cloud base, increasing cloud thickness and variable cloud types) between 2000 LT on 26 December and 2000 LT on 27 December 2017 coincided with the classical picture of preceding upglide clouds of an advancing warm-front system." (please see lines 183-186 in the revised manuscript)

"Furthermore, the radiosonde data exhibited that the southwesterly wind mostly prevailed at the cloud altitudes (Figs. 3d, 3e and 3f), and the air pressure at altitudes of ~0–5 km dropped continuously by ~3–5 hPa in the period (not shown here), which did belong to the typical warm-front features." (please see lines 189–191)

"The air pressure from the radiosonde data at Wuhan showed a persistent decrease (by ~2–4 hPa at altitudes of ~0–5 km) during the observational period from the precursor clouds to precipitation (between 0800 LT on 4 March to 0800 LT on 5 March), that reflected the warm front passage." (please see lines 409–412).

Since the synoptic maps are currently unavailable, the warm-front characteristics are recognized by the temporally-varying cloud properties, decreasing air pressure and southwesterly winds during each of the two cloud/precipitation episodes (please see lines 182-198 and lines 396-412).

*Major comments:*

*2. Verification of LiDAR-measured cloud layer is of importance to the result interpretation, since most of the results presented here are from LiDAR. Given the availability of simultaneously observed radiosonde during both case studies, the authors may make a compare analysis of cloud layers from radiosonde and LiDAR based on the RH threshold methods. This will enhance the readership of this work, in my point of view.*

**Authors' response:**

In light of the reviewer's suggestion, the relative humidity values (over ice) at the lidar cloud base (except cirrus) have been compared with the relative humidity threshold that is conventionally used to determine the cloud base heights.

"At the cloud base (except cirrus), the relative humidity over ice had values close to the relative humidity threshold of 84% that is conventionally used to determine the cloud base heights (Wang and Rossow, 1995; Zhang et al., 2018)." (please see lines 187-189)

In the example of the moderate warm-front rain, it was difficult to determine the relative humidity value at cloud base altitude since the apparent source cloud of the virga was invisible by lidars around radiosonde launch time (due to strong optical attenuation at ~2000 LT on 4 March 2019).

*Minor comments:*

*1. L60-63: "An artificial water splashing.... was altered" reappears in L110-114, which seems redudant and one can be kept.*

**Authors' response:**

The authors sincerely thank the reviewer for his friendly reminding. In the revised manuscript, the sentences (L60-63) in the previous version have turned to "According to an artificial water splashing experiment, water accumulation on the lidar roof windows yielded nearly height-independent lidar signal ($X$, range-corrected signal) attenuation, whereas neither the $X$ vertical structure nor the profile of the volume depolarization ratio $\delta_v$ (the magnitude and vertical structure) were altered." (please see lines 60-63 in the revised manuscript), while the sentences (L110-114) have changed to "An artificial water splashing experiment was performed on the lidar roof windows to examine the effects of water accumulation. A comparison of the lidar profiles with and without water accumulation on the lidar roof windows is given in Figure 1..." (please see lines 128-138), according to another reviewer's suggestion (reviewer 1: showing a plot of the artificial water splashing experiment).

*Minor comments:*

*3. L92: "Based on a method developed by Newsom et al. (Newsom et al., 2009; Zhang et al., 2014), " can be rephrased as "Based on a method originally proposed by Newsom et al. (2009) that was further developed by Zhang et al. (2014),"*

**Authors' response:**

We thank the reviewer for his friendly suggestion. The statement has been rephrased (please see lines 92-94).

*Minor comments:*

*3. L122: "A comparison" -> "A comparison analysis"*

**Authors' response:**

"A comparison" has changed to "A comparison analysis" in the revised manuscript (please see line 147).

*Minor comments:*

*4. L134: "Nash 2011" -> "Nash et al., 2011"*

**Authors' response:**

In the revised manuscript, the "Nash 2011" has been replaced by "Nash et al., 2011" (please see line 159).

*Minor comments:*

*5. L136-137: More details about the measurements by tipping-bucket rain gauge are suggested to be added, such as the sampling intervals or frequency.*

**Authors' response:**

Taking the reviewer's suggestion, more details about the tipping-bucket rain gauge have been added in the revised manuscript.

"It has a sampling interval of 1 min. For each 0.1 mm of precipitation, the bucket tips and empties, yielding an output signal." (please see lines 162-163)

---

## Author Comment (AC3)

**Response to reviewer 3 (RC3, Anonymous Referee #4)**

Reviewer's comments are presented here by italics

***Comments:***
*The paper contains interesting lidar observations of virga below mixed-phase clouds and a detailed, however often speculative interpretation of the virga observations. My overall impression is that the paper in its present form is not in a good shape. The manuscript is also quite long and should be shortened.*

**Authors' response:**

The authors sincerely thank this reviewer for his/her constructive criticisms and valuable suggestions after well-considered reading of our manuscript. Taking all the comments into account, the manuscript has been revised and some parts shortened. Since another reviewer (reviewer 1) suggests that "It might be helpful to make more paragraphs and structure them better. It is not always easy to connect the information with the actual microphysical processes observed. So having more explanation of what process is happening and explain the resulting observation signatures would help.", we have added more (physically reasonable) explanations and paragraphs in the revised manuscript with the ice nucleating processes being not covered.

***Comments:***
*First of all, the lidar setup (zenith pointing, large receiver FOV of 1 mrad) can lead to very low depolarization ratio values introduced by specular reflection (because of zenith pointing). Furthermore, the can sensitively influence the depolarization ratio observations via multiple scattering by droplets. Nothing is mentioned to these instrumental influences. The interesting finding of a local maximum of the depolarization ratio around 600 m height is in the near-range of the lidar so that systematic instrumental effects cannot be excluded. I mean, the overlap profile (incomplete overlap between laser beam and receiver FOV in the near range) is usually not well known and can vary with time (and diurnal cycle). All this causes artefacts. Nothing is mentioned to this problem.*

**Authors' response:**

In terms of our lidar setup, the instrumental effects on the depolarization ratio values (multiple-scattering-induced depolarization enhancement, incomplete overlap between laser beam and receiver FOV in the near range, and an uncertainty of low depolarization ratios in discriminating cloud droplets and falling, oriented ice crystals) have been addressed in the revised manuscript. Please see the following statements:
"Here we can exclude a possibility that the $\delta_v$ maxima (~0.1–0.4) at ~0.6-km altitude resulted from multiple scattering by dense droplets around this altitude. As mentioned above, for the 1-mrad receiver FOV, a dense water-droplet cloud layer with the

multiple-scattering-induced depolarization ratio $\delta_v$ values larger than 0.1 is optically opaque. In contrast to this situation, in our case, when the prominent $\delta_v$ peak (~0.1–0.4) around 0.6-km altitude occurred, the vertical structure of the precipitation streaks at altitudes far above 0.6 km (e.g., ice bright band, lidar dark band and lidar water bright band) was unambiguously detected by our polarization lidar, indicating that the enhanced depolarization ratios around 0.6-km altitude cannot be caused by multiple scattering from dense spherical water droplets therein. Furthermore, since most falling raindrops evaporated and vanished in the liquid-water bright band as indicated by the enhanced water vapor mixing ratio therein and rapidly-decreasing lidar signal on the bottom of the water bright band, small droplets at altitudes below the water bright band were hardly dense enough to generate a strong multiple scattering with $\delta_v \geq 0.1$." (please see lines 259-269 in the revised manuscript)

With a compactly-designed lidar configuration (20-cm Cassegrain telescope), accurate transmitter-receiver alignment and steady lidar environment temperature (with waterproof transparent roof windows), the complete-overlap altitude of our polarization lidar, that the laser-beam receiver-field-of-view (FOV) overlap function becomes unity, is reliably less than 400 m. In fact, the local maximum of the depolarization ratio around 600 m height represents an altitude range from ~1.2 km down to ~0.3 km in which the depolarization ratio values show a precipitation-related enhancement. Therefore, the local $\delta_v$ maximum around 600 m reflects a natural phenomenon rather than an instrumental artefact.

"Such an example is shown in Fig.6. The lidar profiles above the dark band clearly exhibit the typical structure characteristics of a liquid-topped mixed-phase cloud (virga) (see Fig.6 in (Wang and Sassen, 2001)). The mixed-phase cloud top layer was of large $X$ values and very low $\delta_v$ values (~0.01), while lower part of the cloud was characterized by significantly-smaller $X$ values and large $\delta_v$ values (with a maximum up to ~0.33). Furthermore, the cloud top layer had a maximum water vapor mixing ratio $q_v$ and a temperature of ~−8.5 °C (based on radiosonde data at ~2000 LT on 27 December). These observed results suggest that the cloud top layer should be composed of liquid droplets (that was not dense enough to yield detectable multiple scattering), and the lower part of the cloud was mainly precipitating ice crystals (falling ice virga)." (please see lines 318-326)

***Comments:***

*The observations concentrate on the virga zone below the main altocumulus layers. More precise, the investigation is mainly focusing on the part of the virga from the 0°C level towards the surface. Nothing is said, why the ice crystals can survive such a long time at heights below the 0°C height level. Obviously, the melting crystals cool the surrounding air and keep the temperatures close to 0°C, even up to 1 km below the 0°C height level where the radiosonde (obviously ascending outside of virga) indicated temperatures up to 6°C.*

**Authors' response:**

The authors thank the reviewer's friendly suggestion. In the revised manuscript, a sentence that explains why the ice crystals can survive such a long time at heights

below the 0°C height level has been added.

"Long survival time of falling ice crystals at altitudes below the 0°C level might be ascribed to cooling of the surrounding air during their evaporation and melting."(please see lines 173-174)

*Comments:*

*Nothing is mentioned, why raindrops lead to enhanced depolarization ratios! Maybe I overlooked it. Is that because the shapes of the rain drops are no longer spherical during falling, the shape is like the one of pears with the flat side in falling direction…? Please explain!*

**Authors' response:**

The authors appreciate the reviewer's reminding and suggestion, an explanation about why raindrops lead to enhanced depolarization ratios has been inserted in the revised manuscript.

"Here the magnitude and altitude variation of the lidar depolarization ratio $\delta_v$ values allow us to identify where large-sized raindrops form and break up. Falling small-sized raindrops (equivalent diameters ≤ 1.0 mm roughly) are quasi-spherical (Pruppacher and Klett, 1997) and yield small $\delta_v$ values (generally less than 0.1), whereas falling large-sized raindrops (equivalent diameters > 2.8 mm) become nonspherical (with flat or hollow bottom in falling direction) (Pruppacher and Klett, 1997) and lead to large $\delta_v$ values (larger than 0.1)." (please see lines 252-256)

*Comments:*

*Nothing is mentioned about the impact of multiple scattering in layers with high droplet concentration. Maybe the raindrop maximum at 600 m is just caused by multiple scattering by numerous small droplets and not by 'a few big', nonspherical rain drops. In the paper, there are no profiles of particle backscatter and extinction coefficients. So, there is no opportunity to conclude on the multiple scattering effect. I was puzzled by the fact that the depolarization maximum was always around 600 m height, why not at 800 or 400 or at 1000~m height? I speculate that may have to do with systematic instrumental problems in the near-range of the lidar.*

**Authors' response:**

Taking the reviewer's comments into account, the impact of multiple scattering in layers with high droplet concentration has been addressed. With respect to the raindrop $\delta_v$ maximum at ~600 m, we have added the following explanation (considering a fact that the range-corrected signal $X$ is a good proxy for the particle backscatter measure).

"We examined the multiple-scattering-induced depolarization ratio enhancements for an opaque cloud layer composed of dense spherical water droplets by putting a motorized iris on our polarization lidar system. It is indicated that for a receiver FOV of 1 mrad, the enhanced depolarization ratio $\delta_v$ values due to multiple scattering increased from ~0.03 at the $X$ peak altitude to a maximum value of ~0.27 at the weak-signal cutoff altitude with increasing penetration of laser light into the opaque water-droplet cloud layer. Note that for the same receiver FOV (~1 mrad), the

multiple-scattering-induced depolarization ratio $\delta_v$ values were all less than 0.04 within the laser light penetration range in a lightly-dense water-droplet cloud layer (Hu et al., 2006). Combining the earlier multiple-FOV polarization lidar measurements (Hu et al., 2006) and our similar observations yields a suggestion that for the 1-mrad receiver FOV, the multiple-scattering-induced depolarization ratio values larger than 0.10 should result from an opaque water-droplet cloud layer (see Figs. 2 and 4 in (Yi et al., 2021)). In other words, for the 1-mrad receiver FOV, the vertical structure of hydrometeors and aerosols present above a dense water-droplet cloud layer with $\delta_v$ values larger than 0.1 is undetectable by ground-based lidars." (please see lines 118-128)

"Here we can exclude a possibility that the $\delta_v$ maxima (~0.1−0.4) at ~0.6-km altitude resulted from multiple scattering by dense droplets around this altitude. As mentioned above, for the 1-mrad receiver FOV, a dense water-droplet cloud layer with the multiple-scattering-induced depolarization ratio $\delta_v$ values larger than 0.1 is optically opaque. In contrast to this situation, in our cases, when the prominent $\delta_v$ peak (~0.1−0.4) around 0.6-km altitude occurred, the vertical structure of the precipitation streaks at altitudes far above 0.6 km (e.g., ice bright band, lidar dark band and lidar water bright band) was unambiguously detected by our polarization lidar, indicating that the enhanced depolarization ratios around 0.6-km altitude cannot be caused by multiple scattering from dense spherical water droplets therein. Furthermore, since most falling raindrops evaporated and vanished in the liquid-water bright band as indicated by the enhanced water vapor mixing ratio therein and rapidly-decreasing lidar signal on the bottom of the water bright band, small droplets at altitudes below the water bright band were hardly dense enough to generate a strong multiple scattering with $\delta_v \geq 0.1$." (please see lines 259-269)

With respect to the question that the depolarization maximum was always around 600 m height, why not at 800 or 400 or at 1000~m height, further observational and modelling efforts are obviously needed in future. This phenomenon might presumedly reflect a feature for a variety of light mid-level stratiform precipitations. Considering a fact that falling raindrops suffer from strong evaporation during their minutes-long descent in a subsaturated environment, there were no raindrops reaching the surface if no (sparse) large raindrops formed midway. Recently, an optical disdrometer (Parsivel, with 1-min sampling interval) has been installed beside our lidar systems and a newly-developed off-zenith polarization lidar has been placed at our observation site. This allows us in future to detailedly examine the relation between the depolarization maximum around 600-m altitude and precipitation that reached the surface.

With respect to the possible instrumental problems in the near-range of the lidar, please allow us to revisit our previous explanation.

With a compactly-designed lidar configuration (20-cm Cassegrain telescope), accurate transmitter-receiver alignment and steady lidar environment temperature (with waterproof transparent roof windows), the complete-overlap altitude of our polarization lidar, that the laser-beam receiver-field-of-view (FOV) overlap function becomes unity, is reliably less than 400 m. In fact, the local maximum of the depolarization ratio around 600 m height represents an altitude range from ~1.2 km

down to ~0.3 km in which the depolarization ratio values show a precipitation-related enhancement. Therefore, the local $\delta_v$ maximum around 600 m reflects a natural phenomenon rather than an instrumental artefact.

Reference added

Hu, Y., Liu, Z., Winker, D., Vaughan, M., Noel, V., Bissonnette, L., Roy, G., and McGill, M.: Simple relation between lidar multiple scattering and depolarization for water clouds, Opt. Lett., 31(12), 1809–1811, doi:10.1364/OL.31.001809, 2006.

Yi, Y., Yi, F., Liu, F., Zhang, Y., Yu, C., and He, Y.: A prolonged and widespread thin mid-level liquid cloud layer as observed by ground-based lidars, radiosonde and space-borne instruments, Atmos. Res., 263, 105815, https://doi.org/10.1016/j.atmosres.2021.105815, 2021.

***Comments:***

*The most important point of concern is the following: I have a severe problem with the 'theory' of the authors how ice crystals are nucleated via heterogeneous ice nucleation. The authors believe that large (1mm in size) droplets fall out of an altocumulus layer and then they immediately freeze right below cloud base. I have never heart about such an ice nucleation process. Furthermore, I asked myself: How can 1 mm droplets form in an altocumulus cloud layer where typical droplet sizes are 10-20μm ...)*

*The established, common 'theory' of ice nucleation in altocumulus layers is the following:*
*Ice nucleation (dominated by immersion freezing for temperatures >-25°C) starts at the coldest point of the cloud, i.e., at cloud top. At cloud top the probability of ice nucleation is largest because ice nucleation is a strong function of temperature. The probability increases by an order of magnitude when temperature decreases by 5K. Thus in cases of 500-600m thick cloud layers the ice nucleation probability at cloud top is an order of magnitude higher than at cloud base.*

*So, most probably first ice crystals nucleate at cloud top via immersion freezing (liquid water droplets freeze), in the liquid-water droplet environment of the altocumulus layer. Then, in the next step, these ice crystals grow fast and immediately start falling. They grow to about 100μm within 60-120s! During falling they continuously grow as long as ice supersaturation is given. When ice subsaturation levels are reached the ice crystals start to shrink and to evaporate. When entering the air mass below the 0°C height level the crystals start to melt but during this process they consume so much energy that they are able to keep the temperatures of the ambient air close to 0°C (in your cases down to almost 1000m below the 0°C height level) although the radiosonde may have measured 6°C.*

*Finally, the manuscript tells us nothing about the true ice nucleation. There is no information about the altocumulus layer (e.g. cloud top height and temperature), there is no information and discussion about potential seeder/feeder effects (no*

*information about ice cloud layers above the altocumulus), and there is no information about secondary ice formation (triggered by the Hallett Mossop effect) dominating in the height range in which the temperatures are between -5 and -8°C. All this influences the virga properties that are exhaustingly discussed in the paper. It should be clearly said in the introduction that the ice nucleating processes are not covered by the paper. The paper exclusively focus on the virga.*

**Authors' response:**

The authors sincerely appreciate the reviewer's pertinent criticism. The current lidar observations indeed cannot tell anything directly about the true ice nucleation processes. In the revised manuscript, a sentence that "the ice nucleating processes are not covered" has been put into the introduction and appropriate corrections have been made to some earlier wordings.

*Comments:*

*Many aspects mentioned above (but not discussed in the manuscript) triggered many questions! Please do not misunderstand me! I like the approach and want to help to improve the paper. I know that readers appreciate if the authors are critical to their own observations!*

*Major revisions are needed.*

**Authors' response:**

The authors sincerely appreciate this reviewer for his/her great effort in improving the quality of the paper. The manuscript has been revised in light of content in the **Details**.

*Comments:*

*P1, l18: surface rainfall.....? is not just self-explaining: better use: precipitation that reached the surface. You may define 'surface rain' in the introduction, but I personally do not like such wording.*

*P1, l19: parent cloud ...? Is also not self-explaining: better: ...falling out of a shallow mixed-phase cloud layer. Why do we need such a wording? And in the case of seeding from above, then we have grandparent clouds...? I would just call or denote such a clouds ... shallow cloud layer or altocumulus layer.*

*Please change this (surface rain, parent cloud) throughout the article.*

**Authors' response:**

Taking the reviewer's suggestion, the "surface rainfall" changes to the "precipitation that reached the surface" or "reaching-surface precipitation" and the "parent cloud" has been replaced by "a shallow liquid cloud layer" in the revised manuscript. For simplicity of expression, the "apparent source cloud" (e.g., virga and its apparent source cloud) has been used also instead of "parent cloud" in some revised sentences following the wording in an earlier literature ("source cloud" in (Wang and Sassen, 2001)).

A reason for utilizing "a shallow liquid cloud layer" rather than "a shallow mixed-phase cloud layer" is as follows

"Such an example is shown in Fig.6. The lidar profiles above the dark band clearly

exhibit the typical structure characteristics of a liquid-topped mixed-phase cloud (a shallow liquid cloud layer and ice virga below) (see Fig.6 in (Wang and Sassen, 2001)). The mixed-phase cloud top layer (at an altitude of ~4.6 km) was of large $X$ values and very low $\delta_v$ values (~0.01), while lower part of the cloud was characterized by significantly-smaller $X$ values and large $\delta_v$ values (with a maximum up to ~0.33). Furthermore, the cloud top layer had a maximum water vapor mixing ratio $q_v$ and a temperature of ~−8.5 °C (based on radiosonde data at ~2000 LT on 27 December). Combining with the schematic representation of commonly-observed mixed-phase cloud layers (see Fig. 1 in (Bühl et al., 2016)), the current observations suggest that the cloud top layer should mainly be composed of liquid droplets (that were not dense enough to yield detectable multiple scattering), and the lower part of the cloud was mainly precipitating ice crystals (falling ice virga)." (please see lines 318-326)

***Comments:***

*P1, l55-56: I think your hypothesis is wrong: ...suggesting that most supercooled liquid drops falling out of parent clouds rapidly froze into ice crystals on the tops of virga. See my explanation above.*

**Authors' response:**

Taking the reviewer's comments into account, the phrase that "…suggesting that most supercooled liquid drops falling out of parent clouds rapidly froze into ice crystals on the tops of virga." has been replaced by "...indicating that the depolarization ratio values of falling hydrometeors increase rapidly with decreasing altitude on the top of the virgae." (please see lines 55-56)

***Comments:***

*P3,l83: The lidar is pointing exactly vertically! That means you may have very low depolarization ratios from specular reflection by falling, oriented ice crystals. And layers with specular reflection may be misclassified as liquid-droplet layers.*

**Authors' response:**

The authors thank the reviewer for alerting us to the necessary discrimination between liquid-droplet layers and falling, oriented ice crystals when a zenith-pointing polarization lidar has observed very low depolarization ratios. In the revised manuscript, we have inserted the following statements.

"Such an example is shown in Fig.6. The lidar profiles above the dark band clearly exhibit the typical structure characteristics of a liquid-topped mixed-phase cloud (a shallow liquid cloud layer and ice virga below) (see Fig.6 in (Wang and Sassen, 2001)). The mixed-phase cloud top layer (at altitudes of ~4.6 km) was of large $X$ values and very low $\delta_v$ values (~0.01), while lower part of the cloud was characterized by significantly-smaller $X$ values and large $\delta_v$ values (with a maximum up to ~0.33). Furthermore, the cloud top layer had a maximum water vapor mixing ratio $q_v$ and a temperature of ~−8.5 °C (based on radiosonde data at ~2000 LT on 27 December). Combining with the schematic representation of commonly-observed mixed-phase cloud layers (see Fig. 1 in (Bühl et al., 2016)), the current observations suggest that the cloud top layer should mainly be composed of liquid droplets (that were not dense

enough to yield detectable multiple scattering), and the lower part of the cloud was mainly precipitating ice crystals (falling ice virga)." (please see lines 318-326)

*Comments:*
*P3, l84: The receiver field of view is 1 mrad. This means you may have considerable problems with multiple scattering. Multiple scattering in environments with droplets can cause significantly enhanced depolarization ratios. And these high depolarization ratios may be interpreted as ice crystals or as big rain drops. So, there is room for ambiguous interpretation.*
*That should be commented (zenith pointing, specular reflection, multiple scattering, enhanced depolarization, overlap impact, signal gluing impact).*

**Authors' response:**

In order to avoid an ambiguous interpretation on the enhanced depolarization ratios at altitudes around 0.6 km, we have added the following comments in the revised manuscript.

"We examined the multiple-scattering-induced depolarization ratio enhancements for an opaque cloud layer composed of dense spherical water droplets by putting a motorized iris on our polarization lidar system. It is indicated that for a receiver FOV of 1 mrad, the enhanced depolarization ratio $\delta_v$ values due to multiple scattering increased from ~0.03 at the $X$ peak altitude to a maximum value of ~0.27 at the weak-signal cutoff altitude with increasing penetration of laser light into the opaque water-droplet cloud layer. Note that for the same receiver FOV (~1 mrad), the multiple-scattering-induced depolarization ratio $\delta_v$ values were all less than 0.04 within the laser light penetration range in a lightly-dense water-droplet cloud layer (Hu et al., 2006). Combining the earlier multiple-FOV polarization lidar measurements (Hu et al., 2006) and our similar observations yields a suggestion that for the 1-mrad receiver FOV, the multiple-scattering-induced depolarization ratio values larger than 0.10 should result from an opaque water-droplet cloud layer (see Figs. 2 and 4 in (Yi et al., 2021)). In other words, for the 1-mrad receiver FOV, the vertical structure of hydrometeors and aerosols present above a dense water-droplet cloud layer with $\delta_v$ values larger than 0.1 is undetectable by ground-based lidars."(please see lines 118-128)

"Here we can exclude a possibility that the $\delta_v$ maxima (~0.1–0.4) at ~0.6-km altitude resulted from multiple scattering by dense droplets around this altitude. As mentioned above, for the 1-mrad receiver FOV, a dense water-droplet cloud layer with the multiple-scattering-induced depolarization ratio $\delta_v$ values larger than 0.1 is optically opaque. In contrast to this situation, in our case, when the prominent $\delta_v$ peak (~0.1–0.4) around 0.6-km altitude occurred, the vertical structure of the precipitation streaks at altitudes far above 0.6 km (e.g., ice bright band, lidar dark band and lidar water bright band) was unambiguously detected by our polarization lidar, indicating that the enhanced depolarization ratios around 0.6-km altitude cannot be caused by multiple scattering from dense spherical water droplets therein. Furthermore, since most falling raindrops evaporated and vanished in the liquid-water bright band as indicated by the enhanced water vapor mixing ratio therein and rapidly-decreasing

lidar signal on the bottom of the water bright band, small droplets at altitudes below the water bright band were hardly dense enough to generate a strong multiple scattering with $\delta_v \geq 0.1$." (please see lines 259-269)

Added references

Hu, Y., Liu, Z., Winker, D., Vaughan, M., Noel, V., Bissonnette, L., Roy, G., and McGill, M.: Simple relation between lidar multiple scattering and depolarization for water clouds, Opt. Lett., 31(12), 1809–1811, doi:10.1364/OL.31.001809, 2006.

Yi, Y., Yi, F., Liu, F., Zhang, Y., Yu, C., and He, Y.: A prolonged and widespread thin mid-level liquid cloud layer as observed by ground-based lidars, radiosonde and space-borne instruments, Atmos. Res., 263, 105815, https://doi.org/10.1016/j.atmosres.2021.105815, 2021.

The questioning points in brackets have been commented above except "signal gluing impact" that is addressed in the next response.

*Comments:*
*P4, l93: Please state in which height range gluing of signal profiles is performed (for the cases discussed in Sect. 3.)*

**Authors' response:**

The authors sincerely thank the reviewer for his/her reminding. In the revised manuscript, we have added the following statement.

"For the cases in this study, the altitude range of signal gluing was ~1.2–3.3 km. (please see line 94)

*Comments:*
*P4, l104: What about multiple scattering in water-droplet layers, and corresponding increase in depolarization ratios? Please comment on that!*

**Authors' response:**

With respect to the impact of multiple scattering in layers with high droplet concentration, besides the above interpretation, we have further comment as follows:

For a receiver FOV of 1 mrad, the multiple scattering from an optically-thick layer composed of dense spherical water droplets yields initially a monotonic rapid increase in both the lidar signal ($X$) and depolarization ($\delta_v$) with increasing penetration of laser light into the layer as shown in Figure 1 (revised manuscript) and earlier literatures (e.g., Hu et al., 2006). However, the lidar signal $X$ did not show any visible increase with increasing penetration of laser light into the layer that $\delta_v$ value maximized around 0.6-km altitude. Therefore, we can exclude a possibility that the $\delta_v$ maxima at ~0.6-km altitude resulted from multiple scattering by dense droplets around this altitude.

*Comments:*
*P4, l111: Below the dark band, enhanced depolarization ratio indicates rain drops. Please explain why? The shape of rain drops during falling deviates from the perfect spherical form? They look like pears? With the flat surface into the falling direction? Please comment on that.*

**Authors' response:**

Regarding why below the dark band, enhanced depolarization ratio indicates rain drops, the relevant sentences have been revised as follows:

"For some mid-level stratiform precipitations, gravitationally-falling hydrometeors form initially at altitudes above the 0 °C isotherm level. They fall often as mixed-phase hydrometeors (supercooled liquid drops and ice crystals/snowflakes) in sub-zero temperature during their early descent. After the falling mixed-phase hydrometeors pass through the 0 °C isotherm level, the snowflake (ice)-to-raindrop transition can yield a shallow layer of relatively smaller lidar echoes (a local $X$ minimum), that is called "lidar dark band" (Sassen and Chen, 1995; Di Girolamo et al., 2012). The lidar dark band can be used to differentiate between the altitudinal regions with ice-containing particles above the dark band and pure liquid raindrops below the dark band. Hence, at altitudes above the dark band, the discrimination criteria in terms of the depolarization ratio magnitude are $\delta_v < 0.1$ for water droplets/drops and $\delta_v > 0.2$ for ice crystals (Intrieri et al., 2002; Shupe et al., 2008), while an enhanced depolarization ratio ($\delta_v > 0.1$) at altitudes below the dark band indicates the presence of large raindrops." (please see lines 107-116)

With respect to the shape of rain drops during falling, we have added the following statements.

"Falling small-sized raindrops (equivalent diameter $\leq 1.0$ mm) are quasi-spherical (Pruppacher and Klett, 1997) and yield small $\delta_v$ values (generally less than 0.1), whereas falling large-sized raindrops (equivalent diameter $> 2.8$ mm) become nonspherical (with flat or hollow bottom in falling direction) (Pruppacher and Klett, 1997) and lead to large $\delta_v$ values (larger than 0.1)" (please see lines 253-256).

***Comments:***

*P5, l144: An explanation is need why ice crystals can survive as ice crystals over a distance of 600 m below the height of 0°C level, i.e., for about 1200 s (20 min) when falling speed is high with 50 cm/s. To my opinion, cooling of the surrounding air during evaporation and melting… is the reason.*

**Authors' response:**

In light of the reviewer's suggestion, a sentence that explains why the ice crystals can survive such a long time at heights below the 0°C height level has been added.

"Long survival time of falling ice crystals at altitudes below the 0°C level might be ascribed to cooling of the surrounding air during their evaporation and melting."(please see lines 173-174)

***Comments:***

*P6, l 171, 172, 173: parent cloud… I do not like this wording. Furthermore, you do not know whether the crystals were formed in that cloud layer. Maybe ice crystals from above seeded the cloud. So, please be careful with the argumentation. Limit the argumentation to topics and facts that were observed.*

**Authors' response:**

Taking the reviewer's opinion, the "parent cloud" has not been used in the revised manuscript. For simplicity of expression, please allow us to use the "apparent source

cloud" instead of "parent cloud" because mixed-phase hydrometeors (ice virga) were observed falling out of a shallow liquid cloud layer ("apparent source cloud" of ice virga). The "source cloud" came from the same wording in an earlier literature (please see p1674 in (Wang and Sassen, 2001)).

**Comments:**

*P6, Sect 3.3.1: I found this section is too long.*

**Authors' response:**

Taking the reviewer's opinion and some suggestions from another reviewer (RC2, Anonymous Referee #3), Section 3.1.1 has been abbreviated as

"Figure 3 presents the radiosonde profiles that are pertinent to the warm-front cloud at different stages and during precipitation, together with the 1-h mean lidar profiles obtained during the radiosonde launches. The temporally-varying cloud properties (e.g., falling cloud base, increasing cloud thickness and variable cloud types) between 2000 LT on 26 December and 2000 LT on 27 December 2017 coincided with the classical picture of preceding upglide clouds of an advancing warm-front system. Accordingly, a downgoing moist layer was observed strengthening and broadening with time during this period (Figs. 3b and 3c). At the cloud base (except cirrus), the relative humidity over ice had values close to the relative humidity threshold of 84% that is conventionally used to determine the cloud base heights (Wang and Rossow, 1995; Zhang et al., 2018). Furthermore, the radiosonde data exhibited that the southwesterly wind mostly prevailed at the cloud altitudes (Figs. 3d, 3e and 3f), and the air pressure at altitudes of ~0–5 km dropped continuously by ~3–5 hPa in the period (not shown here), which did belong to the typical warm-front features.

The radiosonde released at 0800 LT on 28 December 2017 provided measurements of the meteorological conditions when precipitation reached the surface, although the lidar measurements had already terminated (at 0538 LT) ~ 2 hours earlier. As seen from Figure 3b (red), the relative humidity reached a maximum of 98% with respect to water in an altitude range of ~3–4 km, immediately above the tops of the liquid precipitation streaks (at ~3 km, see Figs. 2a and 2b). Water vapor at altitudes of 3.0–9.0 km was advected from the southwest, as seen in the wind component profiles (Figure 3f, red). The high water vapor mixing ratios observed at altitudes below ~3 km came from the evaporation of falling raindrops." (please see lines 182-198)

**Comments:**

*P7, Sect 3.1.2: Check to what extent multiple scattering effects and specular reflection could have influenced the observations.*

**Authors' response:**

We examined the multiple scattering effects from optically-thick (opaque) cloud layers composed of dense spherical water droplets by using a motorized iris (SID-5714, SmarAct) that was put on our polarization lidar system. An observational example from the multi-field-of-view polarization lidar between 1930 and 1940 LT

on 12 November 2019 is shown in the following Figure R1. Each pair of colored lidar $X$ and $\delta_v$ profiles represent a 1-min integration at a fixed receiver FOV shown in panel (a) with an assumption that the liquid cloud layer was steady during 10-min varying-field-of-view sampling (1930-1940 LT). The altitude resolution is 3.75 m. The assumption was reasonable, because a steady stratiform cloud persistently covered all the sky of our city on this day (12 November 2019) and the air temperature was warmer than 0 °C at altitudes below 4 km according to local radiosonde data. As seen from Figure R1, for a receiver FOV of 1 mrad, the enhanced depolarization ratio due to multiple scattering was less than 0.15 at atitudes where the strong cloud backscatter (enhanced $X$ values) occurred (at altitudes of 1.93–2.06 km). This altitude range (1.93–2.06 km) actually represented the lidar-detectable altitude range (i.e., penetrable range of laser light into the cloud layer). Above the lidar-detectable altitude range, the $X$ value became very weak (one twentieth of its maximum value or less) where the $\delta_v$ value increased from 0.15 at 2.06 km to 0.27 at 2.11 km presumably due to the multiple scattering. This indicated that for a receiver FOV of 1 mrad, the multiple scattering from opaque cloud layers composed of dense spherical water droplets would generate enhanced depolarization ratio values less than 0.27 on their bottom (the lowest part of the opaque cloud layers). In this situation, the lidar is obviously unable to detect cloud layers and/or precipitating hydrometeors occurring at altitudes above the opaque cloud layer. In our cases, when the prominent $\delta_v$ peak (0.1–0.4) around 0.6-km altitude occurred, the vertical structure of the precipitation streaks at altitudes far above 0.6 km (e.g., ice bright band, lidar dark band and lidar water bright band) was unambiguously detected by our polarization lidar, indicating that the enhanced depolarization ratios around 0.6-km altitude cannot be caused by multiple scattering from dense spherical water droplets therein.

[Figure]

Figure R1. Profiles of (a) range-corrected signal $X$ and (b) volume depolarization ratio $\delta_v$ for an opaque stratiform cloud composed of dense spherical water droplets observed by our multi-field-of-view polarization lidar. Each pair of colored lidar $X$

and $\delta_v$ profiles represent a 1-min integration and the altitude resolution is 3.75 m.

With respect to the discrimination between liquid-droplet layers and falling, oriented ice crystals when a zenith-pointing polarization lidar has observed very low depolarization ratios, we have added the following explanation in the revised manuscript.

"The lidar profiles above the dark band clearly exhibit the typical structure characteristics of a liquid-topped mixed-phase cloud (a shallow liquid cloud layer and ice virga below) (see Fig.6 in Wang and Sassen, 2001). The mixed-phase cloud top layer was of large $X$ values and very low $\delta_v$ values (~0.01), while lower part of the cloud was characterized by significantly-smaller $X$ values and large $\delta_v$ values (with a maximum up to ~0.33). Furthermore, the cloud top layer had a maximum water vapor mixing ratio $q_v$ and a temperature of ~−8.5 °C (based on radiosonde data at ~2000 LT on 27 December). Combining with the schematic representation of commonly-observed mixed-phase cloud layers (see Fig. 1 in (Bühl et al., 2016)), the current observations suggest that the cloud top layer should mainly be composed of liquid droplets (that were not dense enough to yield detectable multiple scattering), and the lower part of the cloud was mainly precipitating ice crystals (falling ice virga)." (please see lines 318-326)

***Comments:***

*P8, l218: Why is the volume depol ratio not close to 0.01? Maybe be because of multiple scattering. The paper does not contain any backscatter and extinction coefficients. So, I have no idea whether multiple scattering could be problem or not.*

**Authors' response:**

In the water bright band (please see Fig. 4 in revised manuscript), the volume depolarization ratios of ~0.03−0.06 (not close to 0.01) might result from somewhat nonspherical raindrops that got slightly large sizes during their descent. The multiple scattering from dense droplets would yield a monotonic increase in the volume depolarization ratio $\delta_v$ with increasing penetration of laser light into the water-bright-band layer. However, here the $\delta_v$ value showed a slow decrease with increasing penetration of laser light into the water-bright-band layer. Hence the multiple scattering should not be a problem.

***Comments:***

*P8, l229-230: How can droplets evaporate and at the same time others grow….?*

**Authors' response:**

Taking the reviewer's comment into account, "...the suggestion that most falling pristine raindrops shrunk or vanished in the water bright band due to evaporation, whereas a small portion of them grew to large sizes via collision-coalescence processes and fell out of the water bright band." has changed to "...the suggestion that most falling small-sized raindrops shrunk or vanished in the water bright band due to evaporation, whereas a small portion of large-sized raindrops survived via collision-coalescence processes and fell out of the water bright band." (please see

lines 245-247)

*Comments:*
*P8, l235: ..Peaks at 0.1-0.4? ...you mean 0.1-0.14? Why is the rain depol peak always at 600 m. Can we have an estimate for the backscatter and the extinction coefficient? Maybe multiple scattering had an influence!*
**Authors' response:**
The $\delta_v$ peak values (occurring in Fig. 2 of the revised manuscript) indeed were ~0.1-0.4 (values of ~0.4 occurred at 0536 and 0537 LT on 28 December 2017, the lidar operation terminated exactly at 0539 LT). With respect to the question why is the rain depol peak always at 600 m, further observational and modelling efforts are obviously needed in future. This phenomenon might presumedly reflect a feature for a variety of light mid-level stratiform precipitations. As mentioned above, for a receiver FOV of 1 mrad, the multiple scattering from opaque cloud layers composed of dense spherical water droplets can only result in depolarization ratio values less than 0.27 at altitudes below the weak-signal cutoff altitude. If strong multiple scattering had taken place around 600 m (yielding peak $\delta_v$ values of ~0.1-0.4), i.e., an opaque liquid cloud layer had concealed the atmosphere above 600 m, one would not have observed the vertical structure of the precipitation streaks at altitudes far above 0.6 km (e.g., ice bright band, lidar dark band and lidar water bright band). Hence we can exclude a possibility that the large $\delta_v$ peak values around 600 m are caused by multiple scattering.

*Comments:*
*P9, l240-l244: To my opinion, this is speculation. ...should be avoided.*
**Authors' response:**
Taking the reviewer's opinion, the speculation that "In their further descent, the large raindrops break up into small raindrops, yielding a decrease in the depolarization ratio at altitudes below 0.6 km." has been dropped in the revised manuscript.

*Comments:*
*P9, l269: Large values of the range - corrected signal coinciding with low depol ratio! That could have been caused by specular reflection by a few falling and oriented ice crystals. One should discuss such influences.*
**Authors' response:**
The relationship that the $X$ maxima corresponded to the local minima of the depolarization ratio was detected at altitudes ~600 m below the 0 °C level. However, the specular reflection by falling, oriented ice crystals occurred only at temperatures below –2.5 °C (–2.5 to –40°C) in light of a lidar-observation-based statistics (Westbrook et al., 2010). Furthermore, the altitude positions of the $X$ maxima (local $\delta_v$ minima) showed an irregular variation rather than systematic descent with time. Therefore, we feel difficult to discuss the influences of falling and oriented ice crystals.
Reference:

Westbrook, C.D., Illingworth, A.J., O'Connor, E.J., Hogan, R.J.: Doppler lidar measurements of oriented planar ice crystals falling from supercooled and glaciated layer clouds, *Q. J. R. Meteorol. Soc.,* **136**, 260–276, DOI:10.1002/qj.528, 2010.

***Comments:***

*P9, l279-295: What about seeding by clouds higher up. You have no information about all this. There is so much speculation here. Please avoid that. Keep the discussion short.*

**Authors' response:**

Taking the reviewer's comments into account, the speculation (suggesting that most supercooled liquid drops falling out of their liquid parent cloud rapidly froze into ice crystals) has been dropped. This paragraph has been revised as

"As seen from the $X$ and $\delta_v$ precipitation streaks at altitudes below ~1.5 km (Figs.2a and 2b), precipitation that reached the surface was intermittent. During periods without reaching-surface precipitation, our lidars were able to sample both a complete virga (from the rain to the snow regions) and a shallow mixed-phase cloud layer immediately above the virga under weak optical attenuation conditions. Such an example is shown in Fig.6. The lidar profiles above the dark band clearly exhibit the typical structure characteristics of a liquid-topped mixed-phase cloud (a shallow liquid cloud layer and ice virga below) (see Fig.6 in (Wang and Sassen, 2001)). The mixed-phase cloud top layer (at altitudes of ~4.6 km) was of large $X$ values and very low $\delta_v$ values (~0.01), while lower part of the cloud was characterized by significantly-smaller $X$ values and large $\delta_v$ values (with a maximum up to ~0.33). Furthermore, the cloud top layer had a maximum water vapor mixing ratio $q_v$ and a temperature of ~−8.5 °C (based on radiosonde data at ~2000 LT on 27 December). Combining with the schematic representation of commonly-observed mixed-phase cloud layers (see Fig. 1 in (Bühl et al., 2016)), the current observations suggest that the cloud top layer should mainly be composed of liquid droplets (that were not dense enough to yield detectable multiple scattering), and the lower part of the cloud was mainly precipitating ice crystals (falling ice virga). The liquid-topped mixed-phase cloud (a liquid cloud layer and ice virga below) (Bühl et al., 2016) might be fundamental monomers that constitute mid-level precipitating stratiform clouds. Interestingly, the $\delta_v$ magnitude of the falling virga increased from the liquid-water values of ~0.03–0.10 at an altitude of 4.35 km to the ice/snow values of ~0.21–0.33 at an altitude of 4.0 km. The falling ice crystals yielded a very weak ice bright band at an altitude of ~3.0 km, and then melted into liquid drops at an altitude of ~2.76 km (the local $\delta_v$ minimum). During their further descent, the liquid drops fully vanished due to evaporation, leaving a lidar-detectable rain virga (water bright band) without reaching-surface precipitation. In contrast to the situation during precipitation that reached the surface, no clear-cut $\delta_v$ enhancement occurred at an altitude of approximately 0.6 km when there were only virgae suspended in air. Similar results were discerned for other lidar profiles shown in Figure 2, in which a complete mixed-phase cloud layer could be detected." (please see lines 315-335)

*Comments:*

*P10, l297-308: Here you present again the erroneous ice nucleation theory! You should at least also present the established one (in the absence of seeding from above, ice crystals nucleate at cloud top, grow fast and start falling through the cloudoand become large before they leave the main cloud layer and show up in the virga as quite large crystals that may further grow as long a supersaturation over ice is given.*

**Authors' response:**

Taking the reviewer's comment into account, the ice nucleating processes are not covered in the revised manuscript. The earlier statements (lines 297-308) have changed to

"During the light warm-front rain event, since the reaching-surface precipitations and virgae occurred alternately on a small time scale from a few minutes to tens of minutes and since their precipitation streaks had nearly the same dark-band structures (Figs.2a and 2b), both reaching-surface precipitation and virgae would come from the same source cloud (because a warm-front cloud system is generally widespread and slowly varying). Reaching-surface precipitation (drizzle) arose when the precipitation rate was high below the shallow water-droplet-dominated cloud layer (apparent source cloud), while virgae without reaching-surface precipitation took place when the subcloud precipitation rate was slightly low. Therefore, the current lidar observations reveal the microphysical process of precipitating hydrometeors related to light warm-front rain. Both reaching-surface rainfall and virgae suspended in air began as mixed-phase hydrometeors fell out of a liquid apprent-source cloud layer at altitudes above the 0 °C isotherm level. The depolarization ratio magnitude of falling hydrometeors increased from the liquid-water values ($\delta_v < 0.10$) to the ice/snow values ($\delta_v > 0.25$) during the first 100–200 m of their descent. Subsequently, the falling hydrometeors yielded a dense layer with an ice/snow bright band occurring above and a liquid-water bright band occurring below (separated by a lidar dark band) as a result of crossing the 0°C level." (please see lines 337-348)

*Comments:*

*The entire Sect. 3.1.2 is very long and contains many speculative statements. The entire section should be shortened and should be based on what was observed.*

**Authors' response:**

In the revised manuscript, a sentence that "the ice nucleating processes are not covered" has been put into the introduction (please see line 67) and all relevant speculative statements in Section 3.1.2 have been dropped. Now the content of revised Section 3.1.2 provides important observational information about microphysical process of precipitating hydrometeors from mid-level stratiform clouds and pertinent (physically-reasonable) explanation based on what are observed. Please allow us to keep its present length (revised content) if there is no erroneous statements (Section 3.1.1 has been shortened greatly), because another reviewer (reviewer 1) suggests that "It might be helpful to make more paragraphs and structure them better. It is not always easy to connect the information with the actual microphysical processes observed. So having more explanation of what process is happening and

explain the resulting observation signatures would help."

***Comments:***
*P11, Sect 3.2: Another case, again depol peak caused by rain at 600 m! Why always at 600 m height? Should be clarified and discussed.*

**Authors' response:**

  With respect to the question why the depolarization peak caused by rain was always around 600 m height, further observational and modelling efforts are obviously needed in future. This phenomenon might presumedly reflect a feature for a variety of light mid-level stratiform precipitations. Considering a fact that falling raindrops suffer from strong evaporation during their minutes-long descent in a subsaturated environment, there were no raindrops reaching the surface if no (sparse) large raindrops formed midway. Recently, an optical disdrometer (Parsivel, with 1-min sampling interval) has been installed beside our lidar systems and a newly-developed off-zenith polarization lidar has been placed at our observation site. This allows us in future to detailedly examine the relation between the depolarization maximum around 600-m altitude and precipitation that reached the surface.

***Comments:***
*Speculation about break up processes, occurrence of ensembles of small and large droplets, collision-coalescence effects… all this sounds convincing, but is that the truth? … As long as the role of multiple scattering is not clarified, the discussion is not trustworthy.*

**Authors' response:**

  With respect to the role of multiple scattering, we have added the following statements in the revised manuscript.

"We examined the multiple-scattering-induced depolarization ratio enhancements for an opaque cloud layer composed of dense spherical water droplets by putting a motorized iris on our polarization lidar system. It is indicated that for a receiver FOV of 1 mrad, the enhanced depolarization ratio $\delta_v$ values due to multiple scattering increased from ~0.03 at the $X$ peak altitude to a maximum value of ~0.27 at the weak-signal cutoff altitude with increasing penetration of laser light into the opaque water-droplet cloud layer. Note that for the same receiver FOV (~1 mrad), the multiple-scattering-induced depolarization ratio $\delta_v$ values were all less than 0.04 within the laser light penetration range in a lightly-dense water-droplet cloud layer (Hu et al., 2006). Combining the earlier multiple-FOV polarization lidar measurements (Hu et al., 2006) and our similar observations yields a suggestion that for the 1-mrad receiver FOV, the multiple-scattering-induced depolarization ratio values larger than 0.10 should result from an opaque water-droplet cloud layer (see Figs. 2 and 4 in (Yi et al., 2021)). In other words, for the 1-mrad receiver FOV, the vertical structure of hydrometeors and aerosols present above a dense water-droplet cloud layer with $\delta_v$ values larger than 0.1 is undetectable by ground-based lidars." (please see lines 118-128)

Reference added

Hu, Y., Liu, Z., Winker, D., Vaughan, M., Noel, V., Bissonnette, L., Roy, G., and McGill, M.: Simple relation between lidar multiple scattering and depolarization for water clouds, Opt. Lett., 31(12), 1809–1811, doi:10.1364/OL.31.001809, 2006.

Yi, Y., Yi, F., Liu, F., Zhang, Y., Yu, C., and He, Y.: A prolonged and widespread thin mid-level liquid cloud layer as observed by ground-based lidars, radiosonde and space-borne instruments, Atmos. Res., 263, 105815, https://doi.org/10.1016/j.atmosres.2021.105815, 2021.

"Here we can exclude a possibility that the $\delta_v$ maxima (~0.1−0.4) at ~0.6-km altitude resulted from multiple scattering by dense droplets around this altitude. As mentioned above, for the 1-mrad receiver FOV, a dense water-droplet cloud layer with the multiple-scattering-induced depolarization ratio $\delta_v$ values larger than 0.1 is optically opaque. In contrast to this situation, in our cases, when the prominent $\delta_v$ peak (~0.1−0.4) around 0.6-km altitude occurred, the vertical structure of the precipitation streaks at altitudes far above 0.6 km (e.g., ice bright band, lidar dark band and lidar water bright band) was unambiguously detected by our polarization lidar, indicating that the enhanced depolarization ratios around 0.6-km altitude cannot be caused by multiple scattering from dense spherical water droplets therein. Furthermore, since most falling raindrops evaporated and vanished in the liquid-water bright band as indicated by the enhanced water vapor mixing ratio therein and rapidly-decreasing lidar signal on the bottom of the water bright band, small droplets at altitudes below the water bright band were hardly dense enough to generate a strong multiple scattering with $\delta_v \geq 0.1$." (please see lines 259-269)

***Comments:***
*Sect. 3.2.1: There is no information about the clouds, the ice nucleation processes, potential secondary ice nucleation processes, seeding effects, nothing. That should be emphasized to keep the entire discussion short. You should concentrate on the virga, because more is not possible. That must be clearly stated in the manuscript.*
**Authors' response:**
In light of the previous suggestion of the reviewer, a sentence that "the ice nucleating processes are not covered" has been inserted in the revised Introduction (please see line 67). Therefore, the relevant speculations have been dropped in the Sections 3.2.1-3.2.2. In addition, the statements that "The (apparent) source cloud for this moderate rain event was invisible by the lidars due to strong optical attenuation. Therefore, the following analysis was limited to the ice bright band and below." have been added (please see line 389-390).

***Comments:***
*P12, l369: Ice crystals detected 960m below 0°C height level…. Again, how can they survive? You have to give a reason!*
**Authors' response:**
In light of the previous suggestion of the reviewer, an interpretation that "Such a long survival time of falling ice crystals at altitudes below the 0°C level was due to cooling of the surrounding air during their evaporation and melting." has been added. (please

see lines 419-420)

*Comments:*
*Sect. 3.3.2: ..again, what is the potential impact of multiple scattering?*
**Authors' response:**
 With respect to the impact of multiple scattering, we have added the following interpretation in the revised manuscript.
"As mentioned above, for the 1-mrad receiver FOV, if such large $\delta_v$ values (~0.27–0.35) came from the multiple scattering by a dense water-droplet cloud layer around 0.6-km altitude, the cloud layer would be optically opaque. It would conceal the vertical structure of the precipitation streaks at altitudes above 0.6 km. In contrast to this situation, as seen from Fig. 10, the vertical structure of the precipitation streaks at altitudes above 0.6 km was clearly discerned by our ground-based polarization lidar, indicating that the enhanced depolarization ratios around 0.6-km altitude cannot be caused by multiple scattering from dense spherical water droplets therein. Furthermore, since most falling raindrops evaporated and vanished in the liquid-water bright band as indicated by the enhanced water vapor mixing ratio therein and rapidly-decreasing lidar signal on the bottom of the water bright band, small droplets at altitudes below the water bright band were hardly dense enough to generate a strong multiple scattering with $\delta_v \geq 0.1$. Therefore, it is suggested that the prominent $\delta_v$ peak at an altitude of approximately 0.6 km reflected the collision-coalescence growth of falling large raindrops and their subsequent spontaneous breakup." (please see lines 445-456)

*Comments:*
*Figure 1: The colored figures should be shown from 27 Dec, 00:00 LT to 28 Dec 16 LT, and probably up to 7 km only to see the necessary details. In the present form, the figure is almost useless.*
**Authors' response:**
Figure 1 (i.e., Figure 2 in the revised manuscript) has been replotted with an abscissa from 1600 LT on 26 December to 0600 LT on 28 December 2017 and ordinate from 0 to 12 km. The precipitation streaks have been zoomed in to get a better view of their structure details. We want this figure to exhibit a complete warm-front cloud process and subsequent precipitation based on ground-based lidar observations.

*Comments:*
*Figure 3: Is cloud top at 4 km height? We do not know! The decreased depolarization ratio from 3.6 to 4 km could be caused by specular reflection? Who knows! The decrease of the depolarization ratio below the local maximum at 600 m height may be an instrumental effect (bias) because the observations are performed in the near-range of the lidar where nothing is well defined. Please comment on that! So break up of rain droplets is speculation. Also evaporation could have started and the droplets got smaller and spherical again…*
**Authors' response:**

In the Figure 4 of the revised manuscript (i.e., Figure 3 in the earlier version), the apparent source cloud of falling ice virga is invisible due to optical attenuation. Our description to Figure 4 started from the ice bright band (downward).

Regarding the reliability on the decrease of the depolarization ratio below the local $\delta_v$ maximum at 600 m height, please allow us to briefly introduce our lidar system. With a compactly-designed lidar configuration (20-cm Cassegrain telescope), accurate transmitter-receiver alignment and steady lidar environment temperature (with waterproof transparent roof windows), the complete-overlap altitude of our polarization lidar, that the laser-beam receiver-field-of-view (FOV) overlap function becomes unity, is reliably less than 400 m. Therefore, the depolarization ratio decrease below the local $\delta_v$ maximum at ~600-m altitude is trustworthy at least at altitudes down to 400 m. The break up of raindrops is a possible physical explanation to the $\delta_v$ decrease at altitudes below 600 m. We have noticed that the evaporation effect on the reduction of drop sizes is much more significant for small-sized droplets than large-sized rain drops (because the evaporation reduction rate of the drop size is inversely proportional to the magnitude of drop size). In contrast to the expected evaporation effect, the visible decrease of the depolarization ratios (from > 0.1 to <0.1) occurred in a narrow altitude range (e.g., ~300-600 m, please see Figs. 4b and 5b in the revised manuscript). Therefore, we believe that for large-sized raindrops, the evaporation effect was relatively weaker than the breakup effect.

*Comments:*
*Figure 4: Again the overlap problems in the lowest 500 m! Can we trust the depolarization ratio values at heights below 500 m?*
**Authors' response:**
Please see the interpretation above.

*Comments:*
*Figure 5: Again a layer with low depolarization ratio around 4.5 km height together with large signal! Is that the liquid cloud layer? Or are there falling, oriented crystals producing specular reflection? This time no enhanced depolarization ratio around 600 m height, no big raindrops? This case demonstrates that there is slight decrease of the depolarization ratio from 250 m to 1 km. This is probably the background depol height profile in the absence of any multiple scattering effect.*
**Authors' response:**
With respect to whether the layer with low depolarization ratio around 4.5 km height together with large signal was the liquid cloud layer or falling, oriented crystals yielding specular reflection (please see Figure 6 in the revised manuscript), we have the following explanation

"The lidar profiles above the dark band clearly exhibit the typical structure characteristics of a liquid-topped mixed-phase cloud (a shallow liquid cloud layer and ice virga below) (see Fig.6 in (Wang and Sassen, 2001)). The mixed-phase cloud top layer was of large $X$ values and very low $\delta_v$ values (~0.01), while lower part of the

cloud was characterized by significantly-smaller $X$ values and large $\delta_v$ values (with a maximum up to ~0.33). Furthermore, the cloud top layer had a maximum water vapor mixing ratio $q_v$ and a temperature of ~−8.5 °C (based on radiosonde data at ~2000 LT on 27 December). Combining with the schematic representation of commonly-observed mixed-phase cloud layers (see Fig. 1 in (Bühl et al., 2016)), the current observations suggest that the cloud top layer should mainly be composed of liquid droplets (that were not dense enough to yield detectable multiple scattering), and the lower part of the cloud was mainly precipitating ice crystals (falling ice virga)." (please see lines 318-326). The liquid-topped mixed-phase cloud (a liquid cloud layer and ice virga below) (Bühl et al., 2016) might be fundamental monomers that constitute mid-level precipitating stratiform clouds.

No enhanced depolarization ratio around 600 m altitude corresponded to the period where precipitation did not reach the surface. Considering a fact that falling raindrops suffer from strong evaporation during their minutes-long descent in a subsaturated environment (from an altitude of ~600 m down to the surface), there were no raindrops reaching the surface if no (sparse) large raindrops formed midway. Hence, no enhanced depolarization ratio around 600 m altitude should correspond to no big raindrops there. When there were only virgae suspended in air without surface-reaching precipitation, the $\delta_v$ precipitation streaks should reflect vanishing raindrops due to evaporation.

*Comments:*
*Figure 6: Bad quality, like Figure 1: What do you want to show. There is almost nothing to see!*
**Authors' response:**
Figure 7 (i.e., Figure 6 in the earlier version) has been replotted with the precipitation streaks being zoomed in. We want to show both a complete warm-front cloud process and subsequent precipitation as shown in the lidar profile sequences.

*Comments:*
*Figure 8: Again, one sees the high depolarization ratios indicating ice crystals, one does not see the main cloud layer. One does not know whether the cloud was seeded by higher clouds. One does not see anything. Except the virga of ice crystals, and then the drop in the depolarization ratio, when the crystals melt. Around 600 m again, rain droplets! Always around 600 m! What is the possible reason that the depolarization ratio maximum is always at about 600m. Maybe caused by multiple scattering?*
**Authors' response:**
In the Figure 9 of the revised manuscript (i.e., Figure 8 in the earlier version), the apparent source cloud of falling ice virga is invisible due to optical attenuation. Our description to Figure 9 was limited to altitudes at the ice bright band and below. With respect to the question that the depolarization maximum is always around 600 m height, further observational and modelling efforts are obviously needed in future. This phenomenon might presumedly reflect a feature for a variety of light mid-level stratiform precipitations. Considering a fact that falling raindrops from mid-level

stratiform cloud suffer from strong evaporation during their minutes-long descent in a subsaturated environment, there were no raindrops reaching the surface if no (sparse) large raindrops formed midway.

   With respect to the multiple scattering, we have added the following explanation in the revised manuscript.

"As mentioned above, for the 1-mrad receiver FOV, if such large $\delta_v$ values (~0.27–0.35) came from the multiple scattering by a dense water-droplet cloud layer around 0.6-km altitude, the cloud layer would be optically opaque. It would conceal the vertical structure of the precipitation streaks at altitudes above 0.6 km. In contrast to this situation, as seen from Fig. 10, the vertical structure of the precipitation streaks at altitudes above 0.6 km was clearly discerned by our ground-based polarization lidar, indicating that the enhanced depolarization ratios around 0.6-km altitude cannot be caused by multiple scattering from dense spherical water droplets therein. Furthermore, since most falling raindrops evaporated and vanished in the liquid-water bright band as indicated by the enhanced water vapor mixing ratio therein and rapidly-decreasing lidar signal on the bottom of the water bright band, small droplets at altitudes below the water bright band were hardly dense enough to generate a strong multiple scattering with $\delta_v \geq 0.1$. Therefore, it is suggested that the prominent $\delta_v$ peak at an altitude of approximately 0.6 km reflected the collision-coalescence growth of falling large raindrops and their subsequent spontaneous breakup." (please see lines 445-456)

***Comments:***
*Figure 9 again: Depolarization maximum at 600 m! Why always at 600 m?*
**Authors' response:**

   With respect to the question why the depolarization maximum is always around 600 m height (Figure 10 in the revised manuscript), further observational and modelling efforts are obviously needed in future. This phenomenon might presumedly reflect a feature for a variety of light mid-level stratiform precipitations. Considering a fact that falling raindrops from mid-level stratiform cloud suffer from strong evaporation during their minutes-long descent in a subsaturated environment, there were no raindrops reaching the surface if no (sparse) large raindrops formed midway.

***Comments:***
*Again, please use all of my comments as a constructive contribution to improve the paper. The topic of the article is interesting and deserves publication!*
**Authors' response:**
The authors greatly thank this reviewer for his/her affirmative remark to our article. Truly taking all the comments as constructive opinion and friendly suggestion, we have carefully revised the manuscript.

---

## Author Response (AR2)

**Response to reviewer 3 (2nd round)**

Reviewer's comments are presented here by italics

**Comments:**

**General**

The revised version is significantly better than the submitted one. However, there are still many points that need to be improved and clarified. The discussion and interpretation of the results is based on measured range-corrected signals (attenuated backscatter) and volume depolarization ratio at 355nm (VDR355), rather than on particle backscatter coefficients (BSC355) and particle depolarization ratios (PDR355). These quantities BSC355 and PDR355 would allow a much better and more clear interpretation of the observations.

The use of VDR355 means that any variation in VDR355 may be partly related to (a) changes in the particle-to-molecular backscatter ratio and (b) changes in the ratio of non-depolarizing droplets to depolarizing ice crystals. This ambiguity must be considered in the entire discussion. This ist not the case in the present version of the paper. This has to be improved.

As can be seen from my quite long list of remaining comments and questions, we need another round of revision.

**Authors' response:**

The authors sincerely thank this reviewer for alerting us to the difference between the volume depolarization ratio  $\delta_{\nu}$  /attenuated backscatter coefficient and the particle depolarization ratio  $\delta_{p}$ /particle backscatter coefficient in interpretating the lidar-observed cloud/virga results. In order to clarify this issue, please allow us to examine the functional dependence of  $\delta_{p}$  (PDR355) upon  $\delta_{\nu}$  (VDR355) and upon lidar backscattering ratio R ( $R(z) = \frac{\beta_{a}(z) + \beta_{m}(z)}{\beta_{m}(z)}$ ) based on the polarization lidar measurements from both our current study and earlier literatures. The particle depolarization ratio ( $\delta_{p}$ ) can be obtained from the following equation (Cairo et al., AO, 1999):

$$\delta_p(\mathbf{z}) = \frac{(1+\delta_m)\delta_v(\mathbf{z})R(z) - (1+\delta_v(z))\delta_m}{(1+\delta_m)R(z) - (1+\delta_v(z))},\qquad(1)$$

where  $\delta_m$  is the molecular depolarization ratio. In light of the theoretical calculation

by Behrendt and Nakamura (OE, 2002), the  $\delta_m$  value is ~0.004 for our 0.3-nm bandwidth polarization lidar (355 nm). We have historically measured a minimum depolarization ratio of 0.0067 in clear air as shown by the following Figure R1. The small depolarization excess (0.0027, exceeding the theoretical  $\delta_m$  value of 0.004) can be ascribed to a small remaining ellipticity in the optics or stress birefringence in our waterproof transparent roof windows.

Figure R1. The minimum volume depolarization ratio (VDR, red dotted line in the right panel) in clear air measured by our 355-nm polarization lidar.

Figure R2 illustrates the functional relationship between the derived particle depolarization ratio  $\delta_p$  (from Equation (1)) and lidar-measured volume depolarization ratio  $\delta_v$  for three specified lidar backscattering ratio (R) values. The R values are fixed by considering the earlier lidar measurements. Specifically, the typical values of R for enhanced aerosol load are around 2, for optically thin clouds up to around 10 (please see Lampert et al., ACP, 2010, p2849). The R values were  $\sim 5-8$  on the upper part of evaporating shallow (~400-m thick) ice virgae (please see Figure 4 in (Cheng and Yi, RS, 2020)). Thus, it is reasonable to set a R value of at least 7 for the precipitation-related virgae shown in Figs.4-6 and Figs.9-10 in the revised manuscript because this type of the virga layers is more than 2-km thick with an ice/snow bright band above and a liquid-water bright band below (separated by a lidar dark band). As seen from Figure R2, for the fixed lidar backscatter ratios (R=5,7,10), the functional dependence of  $\delta_p$  (PDR355) on  $\delta_v$  (VDR355) is quasilinear with a zero offset and an apparent slope slightly larger than 1 (the nonlinear term belongs to high-order small quantity) in conventional polarization lidar measurement range ( $\delta_{\nu}$ ~0–0.6). This is a mathematical basis for the  $\delta_{\nu}$  (VDR355) to discriminate whether the dominant lidar backscattering is attributed to spherical or nonspherical particles in a given backscatter volume (or altitude). The  $\delta_p$  magnitude is always slightly larger than the corresponding lidar-measured  $\delta_v$  value with the net increment  $(\delta_p - \delta_v)$  being small in the low- $\delta_{\nu}$ -value range ( $\delta_{\nu} < 0.1$ ), and being relatively large in the high- $\delta_{\nu}$ -value range

(e.g.,  $\delta_v > 0.3$ ). The magnitude of the net increment decreases with increasing *R* value.

Figure R2 enables us to obtain the discrimination criteria of water droplets and ice crystals expressed by the magnitudes of the volume depolarization ratio for a fixed *R* value. If the discrimination criterion of water droplets is defined as  $\delta_p < 0.1$  in terms of the particle depolarization ratio  $\delta_p$  when R = 7, the equivalent criterion is  $\delta_v < 0.09$ in terms of the volume depolarization ratio  $\delta_v$ . If the discrimination criterion of ice crystals is defined as  $\delta_p > 0.2$  in terms of the particle depolarization ratio  $\delta_p$  when R=7, the equivalent criterion is  $\delta_v > 0.17$  in terms of the volume depolarization ratio  $\delta_v$ .

Figure R2. The particle depolarization ratio  $(\delta_p)$  as a function of the lidar-measured volume depolarization ratio  $(\delta_v)$  for specified lidar backscattering ratio (R) values ( $R \ge 7$  is suitable for the precipitation-related virga in this study). The cyan dashed line denotes a linear function with a slope value of 1.0.

In order to further clarify whether the  $\delta_v$  (VDR355) magnitude is valid in discriminating spherical or nonspherical particles in a given backscatter volume (or altitude) for the present study, according to Equation (1), we analyse now the particle depolarization ratio  $\delta_p$  as a function of lidar backscatter ratio *R* for fixed  $\delta_v$  values that are from our current lidar observations.

As the first example in this study, let us see whether the change in the  $\delta_v$  values of falling hydrometeors during the first 100–200 m of their descent coincides with the  $\delta_p$  variation (from the liquid-water values to the ice/snow values). On 28 Dec 2017, the lidar-measured  $\delta_v$  had the mean values of ~0.059 at 4.35-km altitude and ~0.037 at 4.38 km (on the ice virga top, please see Fig.6b in the revised manuscript). Inserting the two  $\delta_v$  values (as parameters) into Equation (1), the  $\delta_p$  (PDR355) is calculated as a function of *R* (plotted in Figure R3). As seen in Figure R3, all the  $\delta_p$  values on the curves for  $\delta_v = 0.059$  and  $\delta_v = 0.037$  are smaller than 0.1 in the entire range of possible *R* values (7-100, the R values should be larger than 7 for the precipitation-related

virga). The maximum  $\delta_p$  values on the curves are respectively 0.069 at 4.35-km altitude and 0.042 at 4.38 km (corresponding to R=7), indicating that the dominant lidar backscattering should be attributed to spherical water drops/droplets at these altitudes.

Figure R3. The  $\delta_p$  (PDR355) as a function of lidar backscatter ratio *R* for the lidar-measured volume depolarization ratio  $\delta_v$  (VDR355) values at two different altitudes (Fig. 6b in the revised manuscript, 28 Dec 2017). The  $\delta_v = 0.059$  and  $\delta_v = 0.037$  are the average values at 4.35-km and 4.38-km altitudes respectively. Note that all the  $\delta_p$  values on the curves for  $\delta_v = 0.059$  and  $\delta_v = 0.037$  (at the ice virga top) are clearly less than 0.1 in the entire range of possible *R* values (7-100).

On the same day (28 Dec 2017), the lidar-measured  $\delta_v$  had the mean values of ~0.220 at 4.17-km altitude and ~0.259 at 4.02 km (at altitudes ~180–330 m below the the ice virga top, please see Fig.6b in the revised manuscript). Inserting the two  $\delta_v$  values (as parameters) into Equation (1), the  $\delta_p$  (PDR355) is calculated as a function of *R* (plotted in Figure R4). As seen in Figure R4, all the  $\delta_p$  values on the curves for  $\delta_v$  =0.220 and  $\delta_v$  =0.259 are larger than 0.2 in the entire range of possible *R* values (7-100, the R values should be larger than 7 for the precipitation-related virga). The minimum  $\delta_p$  values on the curves are respectively 0.223 at 4.17-km altitude and 0.262 at 4.02 km (corresponding to *R*=100), indicating that the dominant lidar backscattering should be attributed to nonspherical ice crystals at these altitudes.

Figure R3 and R4 indicate that the  $\delta_p$  (PDR355) magnitude is mainly controlled by the  $\delta_v$  (VDR355) magnitude (low and high values), and has a very weak dependence on lidar backscatter ratio R (when  $R \ge 7$ ). Hence the conclusion "The depolarization ratio magnitude of falling hydrometeors increased from the liquid-water values ( $\delta_v < 0.09$ ) to the ice/snow values ( $\delta_v > 0.20$ ) during the first 100–200 m of their descent" (P17, 1515) should be valid.